# Fat body phospholipid state dictates hunger-driven feeding behavior

Kevin P Kelly[1†], Mroj Alassaf[1†], Camille E Sullivan[1], Ava E Brent[1], Zachary H Goldberg[1], Michelle E Poling[1], Julien Dubrulle[2], Akhila Rajan[1]*

[1]Basic Sciences Division, Fred Hutch, Seattle, United States; [2]Cellular Imaging Core, Shared Resources, Fred Hutch, Seattle, United States

**Abstract** Diet-induced obesity leads to dysfunctional feeding behavior. However, the precise molecular nodes underlying diet-induced feeding motivation dysregulation are poorly understood. The fruit fly is a simple genetic model system yet displays significant evolutionary conservation to mammalian nutrient sensing and energy balance. Using a longitudinal high-sugar regime in *Drosophila*, we sought to address how diet-induced changes in adipocyte lipid composition regulate feeding behavior. We observed that subjecting adult *Drosophila* to a prolonged high-sugar diet degrades the hunger-driven feeding response. Lipidomics analysis reveals that longitudinal exposure to high-sugar diets significantly alters whole-body phospholipid profiles. By performing a systematic genetic screen for phospholipid enzymes in adult fly adipocytes, we identify Pect as a critical regulator of hunger-driven feeding. Pect is a rate-limiting enzyme in the phosphatidylethanolamine (PE) biosynthesis pathway and the fly ortholog of human PCYT2. We show that disrupting Pect activity only in the *Drosophila* fat cells causes insulin resistance, dysregulated lipoprotein delivery to the brain, and a loss of hunger-driven feeding. Previously human studies have noted a correlation between PCYT2/Pect levels and clinical obesity. Now, our unbiased studies in *Drosophila* provide causative evidence for adipocyte Pect function in metabolic homeostasis. Altogether, we have uncovered that PE phospholipid homeostasis regulates hunger response.

*For correspondence: akhila@fredhutch.org

†These authors contributed equally to this work

Competing interest: The authors declare that no competing interests exist.

## Editor's evaluation

This manuscript posits that genetically-induced misregulation of phospholipid levels in fat cells causes defective hunger-driven feeding behaviors in adult *Drosophila melanogaster*. In parallel, the study also presents the rescue of feeding defects observed in diet-induced obese flies via the overexpression of the rate-limiting enzyme of phospholipid synthesis (PECT) in fly fat. This is an important paper that presents evidence of a potential causative relationship between phospholipid dysregulation and satiety sensing. This work will be of interest to a broad group of metabolism, obesity, and feeding behavior researchers.

## Introduction

Improper hunger-sensing underlies a multitude of eating disorders, including obesity (*Suzuki et al., 2012*). Yet, the cellular and molecular mechanisms governing the breakdown of the hunger-sensing system are poorly understood. In addition to lipid storage, adipocytes play a crucial endocrine role in maintaining energy homeostasis (*Coelho et al., 2013*; *Kuryszko et al., 2016*). Factors secreted by adipocytes impinge on several organs, including the brain, to regulate systemic metabolism and feeding behavior (*Luo and Liu, 2016*; *Chatterjee and Perrimon, 2021*; *Brent and Rajan, 2020*; *Wurfel et al., 2022*). Since lipids play a key role in signaling, adipocyte lipid composition is likely to

regulate hunger perception and feeding behavior. Linking specific changes in adipocyte lipid composition to hunger perception and feeding behavior remains challenging.

While the effects of neutral fat reserves such as triglycerides on feeding behavior have been extensively studied (*Cansell et al., 2014*), less is known about the effects of phospholipids. Phospholipids comprise the lipid bilayer of the plasma membrane and anchor integral membrane proteins, including ion channels and receptors. They are essential components of cellular organelles, lipoproteins, and secretory vesicles (*Shinoda, 2016*). Changes to phospholipid composition can alter the permeability of cell membranes and disrupt intra- and intercellular signaling (*Shinoda, 2016*; *Ben M'barek et al., 2017*; *Sunshine and Iruela-Arispe, 2017*). Numerous clinical studies suggest an association between phospholipid composition and obesity (*Sharma et al., 2013*; *Chang et al., 2019*; *Anjos et al., 2019*). For example, insulin resistance, a hallmark of obesity-induced type 2 diabetes, is strongly associated with alterations in phospholipid composition (*Chang et al., 2019*). Additionally, key phospholipid biosynthesis enzymes are correlated with obesity in human genome-wide association studies (*Sharma et al., 2013*). Despite these intriguing possibilities, a causative link between altered phospholipid composition and metabolic dysfunction is yet to be established. Furthermore, whether altered adipocyte phospholipid composition specifically leads to dysfunctional hunger-sensing is unknown.

Phosphatidylethanolamine (PE) is the second most abundant phospholipid and is essential in membrane fission/fusion events (*Vance, 2015*; *Moon and Jun, 2020*; *Emoto and Umeda, 2000*). PE is synthesized through two main pathways in the endoplasmic reticulum (ER) and the mitochondria (*Farine et al., 2015*). Phosphatidylethanolamine cytidylyltransferase (Pcyt/Pect) is the rate-limiting enzyme of the ER-mediated PE biosynthesis pathway (*Dobrosotskaya et al., 2002*). Global dysregulation in Pcyt/Pect activity has been shown to cause metabolic dysfunction in animal models and humans (*Lim et al., 2011*; *Tsai et al., 2019*). For example, Pyct/Pect deficiency in mice causes a reduction in PE levels, leading to obesity and insulin resistance (*Fullerton et al., 2009*). Similarly, human studies have found that obese individuals with insulin resistance have decreased Pcyt/Pect expression levels (*Yang et al., 2002*). Chronic exposure to a high-fat diet causes upregulation of Pcyt/Pect, associated with increased weight gain and insulin resistance (*de Wit et al., 2008*). These findings suggest that disruptions in Pcyt/Pect activity, and consequently PE homeostasis, are a common underlying feature of obesity and metabolic disorders. What remains largely unknown is whether Pcyt/Pect activity in the adipose tissue directly regulates insulin sensitivity and feeding behavior.

Like humans, chronic overconsumption of a high-sugar diet (HSD) results in insulin resistance, diet-induced obesity (DIO), and metabolic imbalance in flies (*van Dam et al., 2020*; *Smith et al., 2014*; *Arrese and Soulages, 2010*; *Gáliková and Klepsatel, 2018*; *Musselman et al., 2011*; *Walker et al., 2017*). There is deep evolutionary conservation of feeding neural circuits regulating feeding behavior between flies and mammals (*Pool and Scott, 2014*; *Kim et al., 2017*; *Krashes et al., 2009*; *Beshel and Zhong, 2013*; *Wu et al., 2003*), and multiple studies on feeding behavior in *Drosophila* have identified key neurons and receptors involved (*Musso et al., 2019*; *Musso et al., 2021*; *Stanley et al., 2021*; *Lin et al., 2022*; *Chen and Dahanukar, 2017*; *Dus et al., 2015*). Furthermore, like humans, *Drosophila* display altered feeding behavior in response to highly palatable foods (*Kim et al., 2021*; *Tennessen et al., 2014*; *May et al., 2019*; *Small, 2009*; *Volkow et al., 2011*). Additionally, given flies' short lifespan, feeding behavior in response to an obesogenic diet can be monitored throughout the adult fly's lifespan, providing temporal resolution of behavioral changes under DIO (*Ro et al., 2014*; *Pendergast et al., 2017*; *Pendergast et al., 2014*; *Deshpande et al., 2014*; *Post et al., 2019*). Thus, using a chronic HSD feeding regime in adult flies allows for discovering specific mechanisms relevant to human biology.

In this study, we assess the effects of chronic HSD consumption on flies' hunger-driven feeding (HDF) behavior across a 28-day time window. We note that while HSD-fed flies maintain their ability to mobilize fat stores on starvation, they lose their HDF response after 2 weeks of HSD treatment, suggesting an uncoupling of nutrient sensing and feeding behavior. We reveal that changes in phospholipid concentrations in HSD-fed flies occur during HDF loss. We further show that genetic disruption of the key PE biosynthesis enzyme Pect in the fat body, the fly's adipose tissue, results in the loss of HDF even under normal food (NF) conditions. Significantly, Pect overexpression in the fat body is sufficient to protect flies from HSD-induced loss in HDF. Our data suggest that adipocyte PE-phospholipid homeostasis is critical to maintaining insulin sensitivity and regulating hunger response.

## Results

### Exposure of adult *Drosophila* to HSDs for 14 days disrupts HDF

To assess how obesogenic diets alter HDF, we developed a feeding paradigm using a quantitative monitor (Fly Liquid Interaction Counter [FLIC]) to assess feeding activity over time on different diets (*Figure 1A*). In brief, wild-type (w1118) flies were housed in vials containing NF or HSD (30% more sucrose than NF). After exposure to NF or HSD for 5–28 days, flies were subject to a 0% sucrose/agar media (starvation – stv media) for 16 hr to induce hunger. After 16 hr of starvation, we monitored flies' feeding behavior using the FLIC (*Ro et al., 2014*) for 3 hr (see *Figure 1A* and 'Methods'). We observed that NF-stv w1118 flies showed increased feeding compared to flies fed ad libitum (NF-fed) (*Figure 1B*, *Figure 1—figure supplement 1*), which is consistent with an HDF response in vertebrates (*Ellacott et al., 2010*; *McGrath et al., 2019*). However, HSD-fed flies, though they displayed an HDF response on days 3–10 of HSD exposure, showed a progressive loss of the HDF response to starvation starting on day 14 (*Figure 1C*, shaded in yellow, and *Figure 1—figure supplement 1A*). Excluding the possibility that the loss in HDF is due to increased baseline feeding on HSD, flies on NF and HSD showed similar ad libitum feeding behavior (*Figure 1D*), suggesting that beyond day 14, HSD hunger-sensing is altered.

We then wondered whether HSD-fed flies were capable of sensing and responding to starvation by breaking down energy reserves. Triacylglycerides (TAGs) are the largest energy reserve and are mobilized when flies are starved (*Heier and Kühnlein, 2018*). Thus, changes in TAG levels following starvation can be used as a readout for cellular energy-sensing in *Drosophila* (*Hildebrandt et al., 2011*; *Heier et al., 2021*). We observed that HSD-fed flies displayed similar levels of starvation-induced TAG breakdown to NF-fed flies throughout the 4-week window (*Figure 1E–G*), suggesting that starvation-induced lipolysis remains functional in HSD-fed flies. Even though starvation-induced TAG mobilization in HSD-fed flies is intact throughout the 4-week time window (*Figure 1E–G*), the HDF response only dampens after 14 days of HSD (*Figure 1C*). Overall, this suggested that while flies maintain HDF up to 10 days of HSD exposure, starting at 14 days of HSD, there is a loss of hunger response. However, an alternative hypothesis is that fatter flies (*Figure 1—figure supplement 2*) have delayed hunger perception due to the extra levels of stored energy and, therefore, a delayed need for food and would need to be starved longer to become hungry. To test this, 14-day HSD-fed flies were starved for a range of fasting times (16–32 hr) and then assayed for HDF. We observed that at 16, 24, and 32 hr of starvation, flies do not display HDF (*Figure 1H*). At 24 and 32 hr of starvation, HSD-fed flies broke down 50% of their fat stores (*Figure 1I and J*). At 24 and 32 hr, flies do not display increased feeding in response to this significant energy deficit (*Figure 1H–J*). Notably, although 5–10-day HSD-fed flies have comparable TAG levels to 14-day HSD-fed flies (*Figure 1F*), they display a robust hunger response within 16 hr of starvation (*Figure 1C*, *Figure 1—figure supplement 1*). In contrast, at 14 days of HSD exposure, even at 32 hr of starvation, when flies break down >50% of TAG stores (*Figure 1I and J*), they do not display a robust and sustained hunger response (*Figure 1H*). Hence, subjecting flies to a 14-day HSD disrupts their HDF regardless of TAG stores and fasting time.

### Insulin resistance and lipid morphology changes at the point of HDF loss

The loss of HDF behavior after 14 days of HSD suggests that this is a critical timepoint of metabolic disruption. Given that insulin signaling is a major regulator of systemic metabolism (*Kleinridders et al., 2014*; *Petersen and Shulman, 2018*) and insulin resistance is a hallmark of obesity (*Wu and Ballantyne, 2020*), we asked whether HSD-fed flies display dysfunctional insulin signaling. To answer this, we first measured the amount of *Drosophila* insulin-like peptide 5 (Dilp5) accumulation in the brain's insulin-producing cells (IPCs). It has been shown that Dilp5 accumulation in the IPCs directly correlates with nutritional status (*Géminard et al., 2009*); when flies experience a nutrient-rich environment, Dilp5 levels in the IPCs drop due to increased insulin secretion (*Pasco and Léopold, 2012*). Consistent with the previous reports (*Pasco and Léopold, 2012*), we found that 14 days of HSD feeding ad libitum resulted in decreased Dilp5 accumulation in the IPCs (*Figure 2A*). To determine the downregulation of Dilps at the transcript level, we analyzed the expression of the IPC-secreted Dilps 2 and 5 (*Rulifson et al., 2002*). We found that 14 days of HSD treatment did not alter the expression of Dilps 2 and 5 (*Figure 2—figure supplement 1*). These results (*Figure 2A*, *Figure 2—figure supplement 1*) suggest that HSD increases Dilp5 secretion.

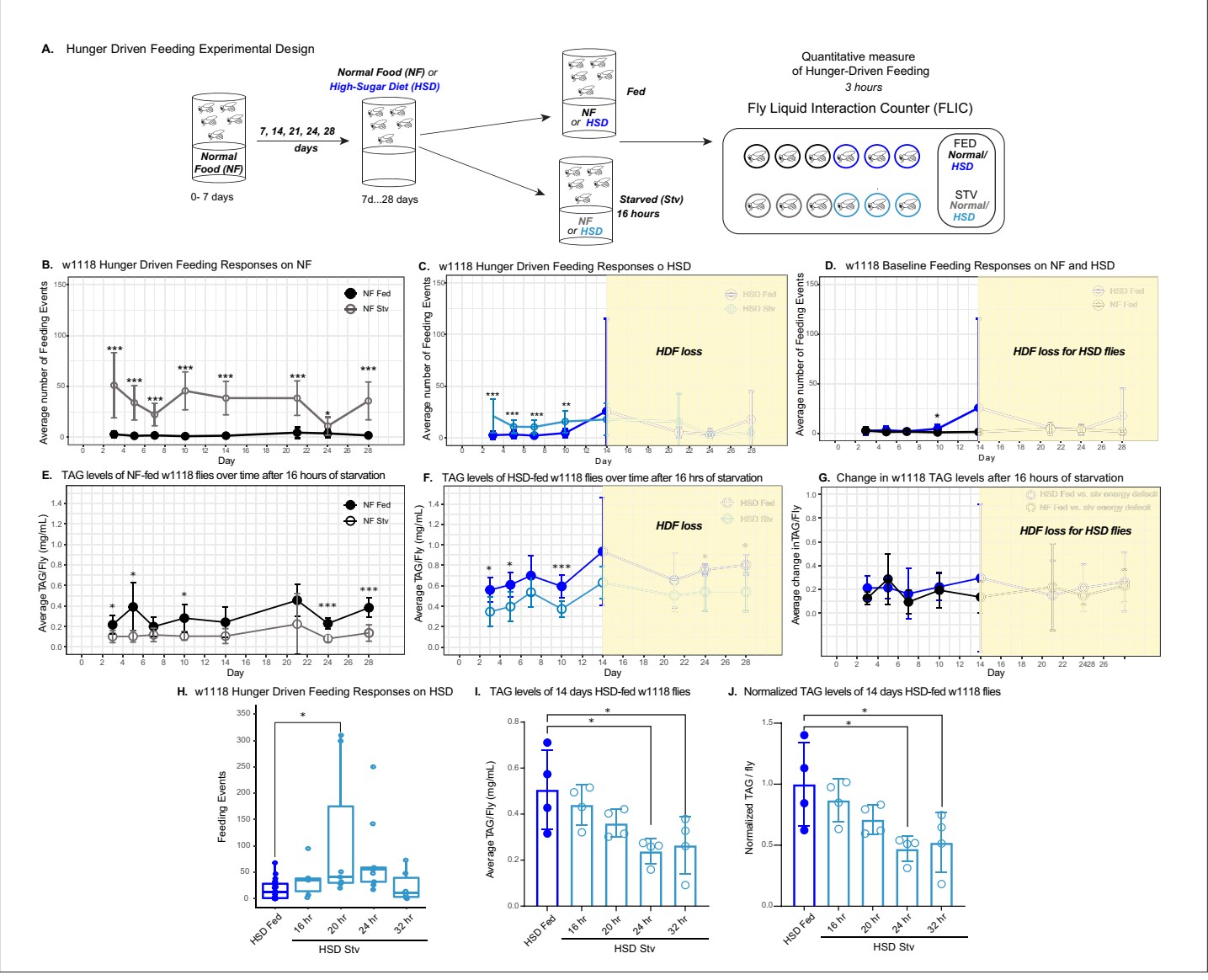

**Figure 1.** High-sugar diet (HSD) causes progressive loss in hunger-driven feeding (HDF). (**A**) HDF behavior in flies was tested using the schematic in (**A**). After aging flies for 7 days on normal lab food, flies we subject to a normal diet or an HSD (30% more sugar in food) for a duration of 3–28 days. For every timepoint, hunger was induced by subjecting flies to starvation (agarose, 0% sucrose media) overnight for 16 hr. Quantitative feeding behavior was monitored using the Fly Liquid Interaction Counter (FLIC) during a 3 hr window immediately after the starvation period. (**B**) Average feeding events over time for normal food (NF) flies that were either fed (filled circles) or starved for 16 hr prior to measurement (stv; no fill). Note that HDF is maintained throughout the experiment. (**C**) Average feeding events over time for HSD-fed flies that were either fed or starved for 16 hr prior to measurement (stv). Note the loss of HDF on day 14. (**D**) Comparison of basal feeding in NF and HSD fed flies over time. N = 18 for each treatment and timepoint. (**E, F**) Average triacylglyceride (TAG) levels per fly of NF (**E**) and HSD-fed (**F**) flies at baseline and after 16 hr of starvation. (**G**) Average change in TAG levels after starvation in NF and HSD flies (n = 8–9 for each treatment and timepoint). (**H**) Flies given HSD for 14 days were starved for various durations on 0% sucrose media (16, 20, 24, and 32 hr, respectively) with feeding events recorded as in (**B–D**) (n = 6–20 for each treatment). Starvation periods were staggered such that all flies were assayed on the FLIC at the same time. (**I**) Average TAG levels per fly of (**H**) flies after subjected to starvation of various durations (n = 4 for each treatment). (**J**) TAG levels per fly of (**H**) flies normalized to the HSD fed control. Two-way ANOVA with Sidak post-test correction. Asterisks indicate significant changes with p-value<0.05, p-value<0.005, and p-value<0.0005. Error bars = standard deviation.

The online version of this article includes the following source data and figure supplement(s) for figure 1:

**Source data 1.** w1118 hunger-driven feeding responses related to *Figure 1B–D*.

**Source data 2.** Change in w1118 triacylglyceride (TAG) levels after 16 hr of starvation related to *Figure 1E–G*.

**Source data 3.** w1118 hunger-driven feeding responses on high-sugar diet (HSD) related to *Figure 1H*.

*Figure 1 continued on next page*

*Figure 1 continued*

**Source data 4.** Triacylglyceride (TAG) levels of 14-day high-sugar diet (HSD)-fed w1118 flies related to *Figure 1I*.

**Source data 5.** Normalized triacylglyceride (TAG) levels of 14-day high-sugar diet (HSD)-fed w1118 flies related to *Figure 1J*.

**Figure supplement 1.** Flies on high-sugar diet (HSD) show a progressive loss in hunger-driven feeding (HDF).

**Figure supplement 1—source data 1.** w1118 hunger-driven feeding responses on normal food (NF) and high-sugar diet (HSD) related to *Figure 1— figure supplement 1A*.

**Figure supplement 2.** Normal food (NF) and high-sugar diet (HSD) flies show similar rates of triacylglyceride (TAG) breakdown following a starvation challenge.

**Figure supplement 2—source data 1.** Triacylglyceride (TAG) levels of normal food (NF) and high-sugar diet (HSD)-fed w1118 flies over time at baseline.

We reasoned that, as previously reported (*Musselman et al., 2011*), the changes in insulin signaling may lead to insulin resistance in peripheral tissue. To address this, we measured the fat body's forkhead box O (FOXO) nuclear localization (*Figure 2B*). Insulin signaling is activated by insulin binding to its cell surface receptor (IR) (*Brogiolo et al., 2001*; *Böhni et al., 1999*). When insulin activates IR, it triggers a phosphorylation cascade of multiple downstream targets, including the transcription factor FOXO (*Kramer et al., 2003*; *Teleman et al., 2005*). Phosphorylation of FOXO prevents it from entering the nucleus and initiating the gluconeogenic pathway, a starvation response pathway. Thus, FOXO nuclear localization has been used as a proxy to monitor insulin sensitivity (*Lee and Dong, 2017*; *Gross et al., 2008*). Since the fat body is a major target for insulin signaling (*Tain et al., 2021*), we wondered whether increased insulin signaling in HSD led to insulin resistance. Indeed, we found that 14 days of HSD treatment resulted in elevated nuclear FOXO levels compared to the NF treatment. Notably, this effect was not seen after acute 6-hr exposure to HSD (*Figure 2—figure supplement 2*).

Accumulation and enlargement of lipid droplets, the cell's lipid storage organelles, are associated with insulin resistance (*Kim et al., 2015*). Flies on HSD showed progressively larger, misshapen, and denser lipid droplets than flies on NF (*Figure 2C*). Thus chronic 14-day HSD treatment reduces Dilp5 in the IPCs, increases nuclear localization of FOXO in the fat body, and alters lipid droplet morphology; all these changes are consistent with the onset of insulin resistance.

## Lipidomics on HSD uncovers alterations in whole-body phospholipid levels at 14-day HSD

Next, we sought to characterize the progressive changes in the lipidome of flies subjected to HSD using a targeted quantitative lipidyzer (see 'Methods'). Seven-day exposure to HSD did not significantly change the overall concentrations of lipid classes (*Figure 3*, *Figure 3—figure supplement 1*). But a prolonged 14-day exposure significantly increased TAGs and diacylglycerides (DAGs) (Figure S4A). At the same time, free fatty acids (FFAs) were surprisingly lower in HSD-fed flies than in NF (see 'Discussion'). Intriguingly, prolonged 14-day exposure to HSD caused a significant increase in the PE and phosphatidylcholine (PC) levels (*Figure 3*). We also observed that two other minor phospholipid classes – lysophosphatidylcholine (LPC) and lysophosphatidylethanolamine (LPE)— are downregulated on 14-day HSD. In particular, LPE downregulation was statistically significant (*Figure 3*). We speculate that LPE reduction likely affects upregulated PE synthesis as LPE serves as a precursor for PE (*Riekhof and Voelker, 2006*; see 'Discussion'). These findings indicate that altered phospholipid levels correlate with HDF loss and insulin resistance.

## Fat body Pect levels regulate apolipoprotein delivery to the brain

PC and PE are primarily synthesized by the fat body and trafficked in lipophorin particles (Lpps) chaperoned by ApoLpp to other organs, including the brain (*Palm et al., 2012*). ApoLpp is the functional ortholog of human ApoB (*Palm et al., 2012*). In flies, Apolpp is post-translationally cleaved into ApoI and ApoII. ApoII is the fragment of Apolpp that harbors the lipid-binding domain. PE-rich ApoII particles have been shown to traffic from fat to the brain (*Palm et al., 2012*). Since PE levels are elevated after 14 days of HSD treatment (*Figure 3*), we predicted increased ApoII levels in the brain. Using an antibody generated against ApoII (*Figure 4—figure supplement 1*), we asked whether 14 days of HSD treatment would increase ApoII delivery to the brain. Surprisingly, we found that though HSD

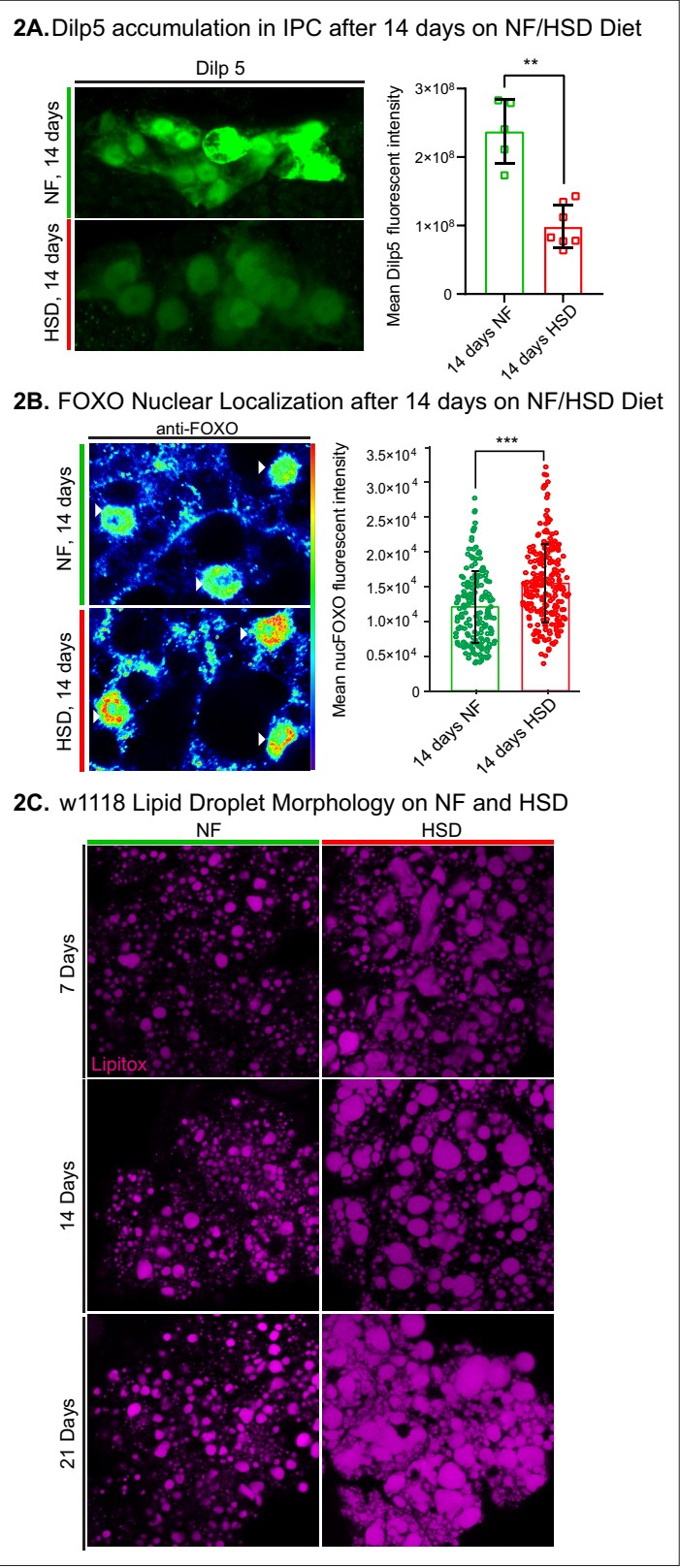

**Figure 2.** *Drosophila* insulin-like peptide 5 (DILP5)/forkhead box O (FOXO) accumulation and lipid droplet morphology altered under high-sugar diet (HSD) exposure. (**A**, left) Representative confocal images of Dilp5 accumulation in insulin-producing cells (IPCs) of normal food (NF) flies (top panel) and HSD-fed flies on day 14 (bottom panel). (**A**, right) Mean Dilp5 fluorescent intensity from z-stack summation projections of IPCs from NF

*Figure 2 continued*

and HSD-fed flies. N = each square represents a single fly. All data were collected from a single experiment. Unpaired *t*-test with Welch's correction. (**B**) Representative confocal images of nuclear FOXO accumulation in the fat bodies of NF flies (top panel) and HSD-fed flies on day 14 (bottom panel). The nuclei are marked with anti-lamin (magenta). Arrowheads point to nuclei. (**B**, right) Mean nuclear FOXO fluorescent intensity from z-stack summation projections of fat bodies from NF and HSD-fed flies. N = each circle represents a nucleus. All data were collected from a single experiment. Two-sided Wilcoxon rank-sum test. Error bars = standard deviation. Asterisks indicate significant changes with p-value<0.05, p-value<0.005, and p-value<0.0005. (**C**) Representative confocal images of lipid droplets (magenta) across time in the fat bodies of NF and HSD-fed flies.

The online version of this article includes the following source data and figure supplement(s) for figure 2:

**Source data 1.** *Drosophila* insulin-like peptide 5 (Dilp5) accumulation in insulin-producing cell (IPC) after 14 days on normal food (NF)/high-sugar diet (HSD).

**Source data 2.** Forkhead box O (FOXO) accumulation in w1118 (14 days after diet treatment).

**Figure supplement 1.** Chronic high-sugar diet (HSD) treatment does not affect *Drosophila* insulin-like peptides (Dilps) 2 and 5 transcript levels.

**Figure supplement 1—source data 1.** *Drosophila* insulin-like peptide (Dilp) 2 and 5 mRNA expression levels in w1118.

**Figure supplement 2.** Acute high-sugar diet (HSD) exposure does not affect forkhead box O (FOXO) nuclear localization.

**Figure supplement 2—source data 1.** Forkhead box O (FOXO) accumulation in w1118 (6 hr after diet treatment).

treatment caused an increase in the overall PC and PE lipid levels (***Figure 3***), the amount of ApoII that chaperones PE to the brain was significantly reduced in an area of the brain proximal to the IPCs (***Figure 4A***). This observation suggests that fat-to-brain trafficking of PE via ApoII particles is disrupted after 14 days of HSD (see 'Discussion).

ApoLpp is synthesized exclusively by the fat body, and Lpps are enriched in PE (***Palm et al., 2012***). Hence, we hypothesized that altering the PE biosynthesis pathway genetically, via RNAi-mediated knockdown, of the crucial enzymes Pect and easily shocked (eas) (***Lim et al., 2011***; ***Tsai et al., 2019***; ***Nyako et al., 2001***) would affect ApoII levels in the brain. Using qPCR, we validated the efficiency of Pect knockdown (***Figure 6—figure supplement 1A***). We found that while eas did not alter the levels of ApoII in the brain, Pect, the rate-limiting enzyme of PE biosynthesis, caused a reduction in ApoII levels like that of HSD (***Figure 4B***). Conversely, fat body-specific Pect overexpression increased ApoII levels in the brain (***Figure 4C***). Given that HSD and Pect knockdown caused a similar effect on ApoII levels in the brain, we predicted that Pect mRNA levels would be low in the HSD-fed flies. To our surprise, we found that 14 days of HSD caused a 200- to 270-fold rise in Pect mRNA levels that fell sharply by day 21 (***Figure 6—figure supplement 1B***) compared to the modest ~7-fold increase in the Pect OE flies (***Figure 6—figure supplement 1A***, right). Hence, extreme deregulation of Pect levels, either up or down, may affect PE levels and their carrier ApoII (see 'Discussion'). This interpretation is consistent with previous studies linking Pect dysregulation to abnormal lipid metabolism and signaling (***Fullerton et al., 2009***; ***Yang et al., 2002***; ***de Wit et al., 2008***).

## Fat body-specific Pect knockdown shows FOXO accumulation and lipid morphology changes similar to HSD

Given that PE homeostasis is disrupted under HSD (***Figure 3***), which coincides with signs of insulin resistance in the fat body (***Figure 2B and C***), we asked whether fat body-specific knockdown of Pect affects insulin sensitivity. Therefore, we measured the nuclear FOXO levels of fat body-specific Pect knockdown flies (***Figure 5A***). We found that, like the HSD-treated flies (***Figure 2B***), Pect knockdown flies exhibit increased nuclear localization of FOXO. Since lipid droplets serve as a repository for lipids (***Yu and Li, 2017***), the building blocks of phospholipids, we reasoned that knocking down Pect will lead to excessive accumulation of lipids and result in enlarged lipid droplets resembling flies on HSD. Indeed, a qualitative view of lipid droplets in Pect knockdown flies showed larger and more clustered lipid droplets than in control (***Figure 5B***). These data suggest that Pect is important for mediating insulin sensitivity in the fat body, and its loss leads to DIO-like phenotypes.

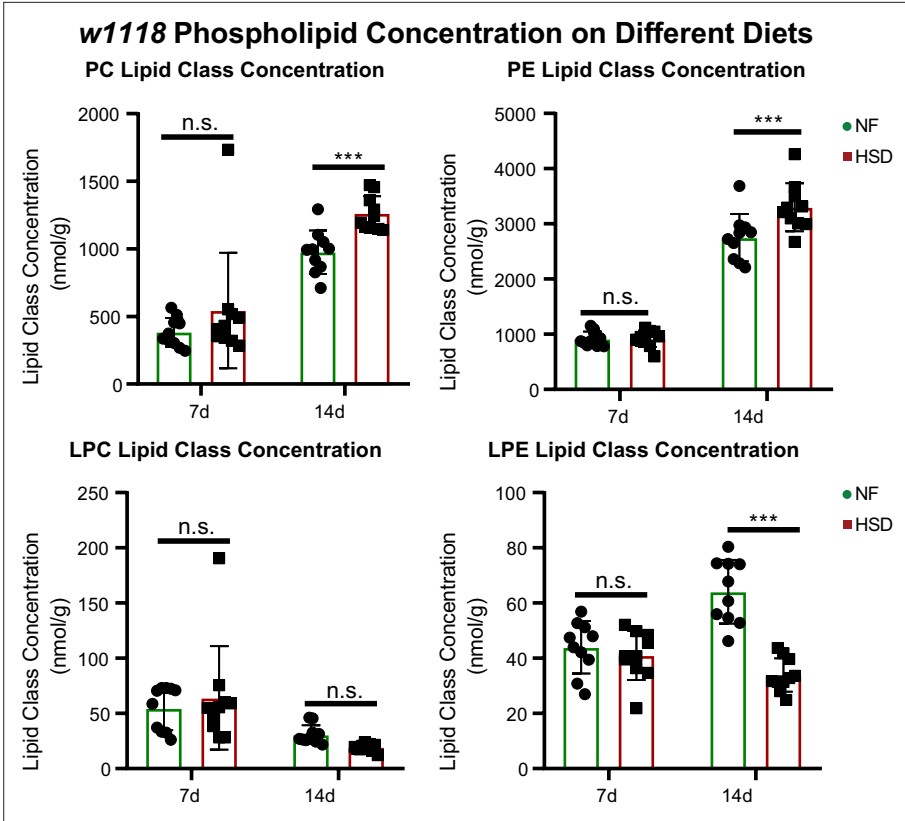

**Figure 3.** Phospholipids are elevated in whole fly during extended high-sugar diet (HSD) feeding. Average concentrations of phosphatidylcholine (PC), phosphatidylethanolamine (PE), lysophosphatidylcholine (LPC), and lysophosphatidylethanolamine (LPE) in flies subjected to normal food (NF) (green) or an HSD (red) for 14 days. Lipidomics was performed using a targeted quantitative lipidyzer (Sciex 5500 Lipidyzer). Ten independent biological replicates were used for each diet and each day, with n = 10 flies composing one biological replicate. Two-way ANOVA and Sidak post-test correction. Asterisks indicate significant changes with p-value<0.05, p-value<0.005, and p-value<0.0005. Error bars = standard deviation.

The online version of this article includes the following source data and figure supplement(s) for figure 3:

**Source data 1.** Lipidomics data for lipid classes of normal food (NF) and high-sugar diet (HSD)-fed flies on day 7 presented in *Figure 3* and *Figure 3—figure supplement 1*.

**Source data 2.** Lipidomics data for lipid classes of normal food (NF) and high-sugar diet (HSD)-fed flies on day 14 presented in *Figure 3* and *Figure 3—figure supplement 1*.

**Figure supplement 1.** High-sugar diet (HSD) alters the lipidome and increases the concentration of phosphatidylethanolamine (PE) and phosphatidylcholine (PC) double bond species.

## Pect knockdown in the fat body alters whole-body phospholipid concentrations

Next, we sought to determine the impact of Pect on the lipidome of adult flies. We compared the lipidomic profile of whole flies expressing Pect-RNAi using a fat-specific driver (Lpp-Gal4) against a control-RNAi (luciferase-RNAi; *Figure 6*). We found that knocking down Pect in the fat body slightly reduced whole-body PE levels (*Figure 6A*). Our results are consistent with a previous study showing that Pect null mutants did not display significant alterations in the levels of PC and PE (*Tsai et al., 2019*). Again, consistent with the same study (*Tsai et al., 2019*), we observed that specific PE species showed significant alterations, with PE 36.2 displaying a significant downregulation (*Figure 6B*; see 'Discussion').

Furthermore, while other major lipid classes did not significantly deviate from control (*Figure 6— figure supplement 3*), we noted a striking and significant increase in two minor classes of phospholipids, LPC and LPE (*Figure 6A*). Given that LPE can serve as a precursor for PE via the exogenous

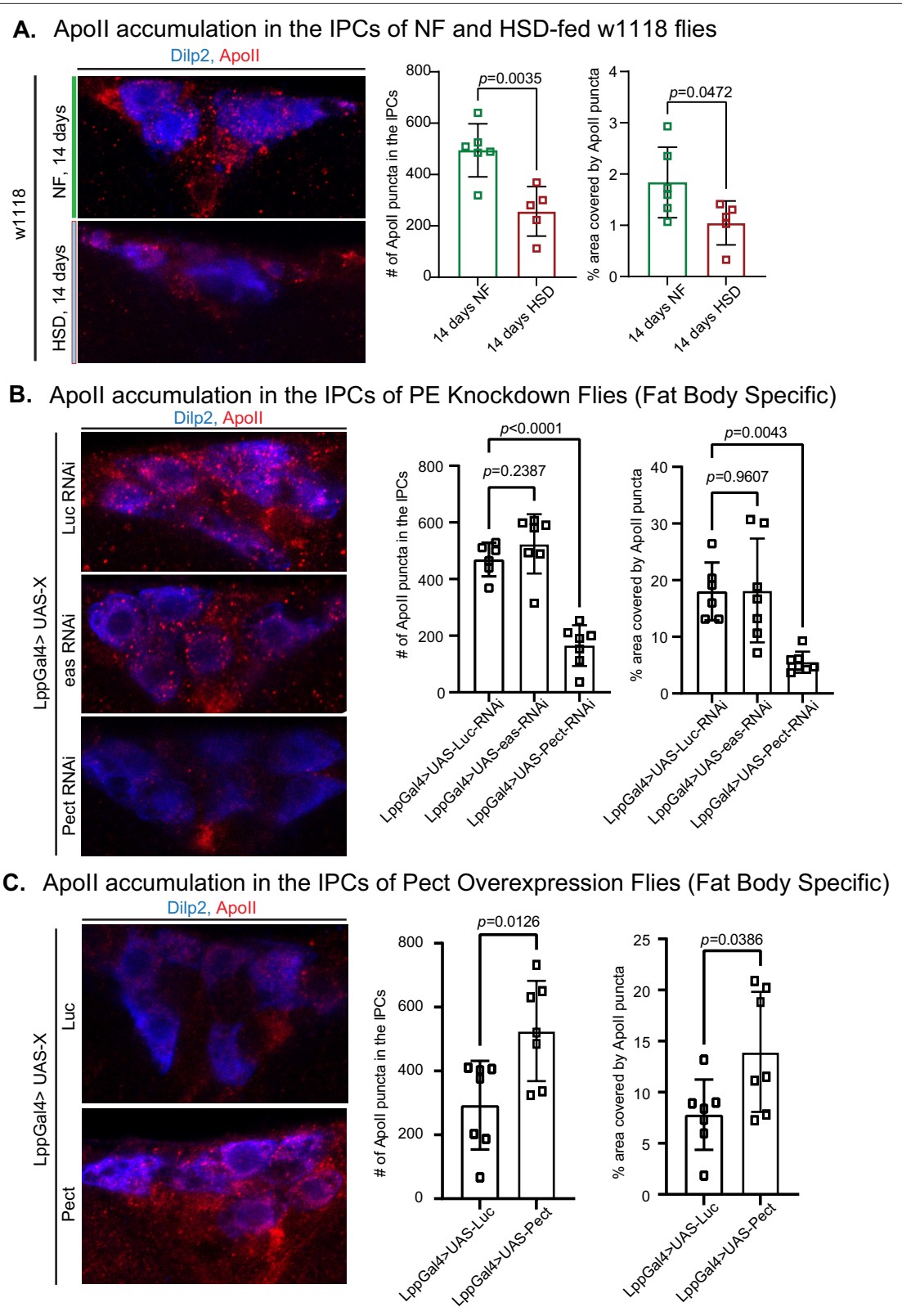

**A.** ApoII accumulation in the IPCs of NF and HSD-fed w1118 flies

**B.** ApoII accumulation in the IPCs of PE Knockdown Flies (Fat Body Specific)

**C.** ApoII accumulation in the IPCs of Pect Overexpression Flies (Fat Body Specific)

**Figure 4.** High-sugar diet (HSD) and knockdown of Pect lead to decreased ApoII levels in the brain. (**A–C**, left) Representative confocal images of ApoII (red) levels in the insulin-producing cells (IPCs) (blue) of (**A**) normal food (NF) and HSD-fed flies on day 14, (**B**) flies with a fat-specific knockdown of Luc, eas, and Pect, and (**C**) flies with a fat-specific overexpression of Luc and Pect. (**A–C**, right) Mean number of ApoII puncta and % area covered by ApoII puncta in the IPCs of (**A**) NF and HSD-fed flies on day 14, (**B**) flies with a fat-specific knockdown of Luc, eas, and Pect, and (**C**) flies with fat-specific

*Figure 4 continued on next page*

*Figure 4 continued*

overexpression of Luc and Pect. N = each square represents a single fly. All data were collected from a single experiment. Unpaired *t*-test with Welch's correction. Asterisks indicate significant changes with p-value<0.05, p-value<0.005, and p-value<0.0005. Error bars = standard deviation.

The online version of this article includes the following source data and figure supplement(s) for figure 4:

**Source data 1.** ApoII accumulation in the insulin-producing cells (IPCs) of normal food (NF) and high-sugar diet (HSD)-fed w1118 flies.

**Source data 2.** ApoII accumulation in the insulin-producing cells (IPCs) of phosphatidylethanolamine (PE) knockdown flies.

**Source data 3.** ApoII accumulation in the insulin-producing cells (IPCs) of Pect overexpression flies.

**Figure supplement 1.** Validation of ApoII antibody.

lysolipid metabolism (ELM) pathway (*Riekhof and Voelker, 2006*), we speculate that in the absence of the rate-limiting enzyme for PE synthesis, there is an elevation in LPE levels (see 'Discussion'). We also performed the same analysis in a Pect overexpression fly line but found no change in any lipid classes compared to the control (*Figure 6—figure supplement 4*). Together, this suggests that fat-specific knockdown of Pect is sufficient to cause a reduction in certain PE species and upregulation of the minor phospholipid classes, especially the PE precursor class of minor phospholipid LPE. Hence, we conclude that fat body-specific Pect disruption results in systemic phospholipid class composition imbalance.

## Pect knockdown in the fat modulates HDF behavior

We next sought to determine whether phospholipid imbalance can impact feeding behavior. Given that the fat body is a major source of phospholipids synthesis (*Palm et al., 2012*), we asked whether disrupting PE and PC synthesis would impact HDF behavior in NF and HSD feeding conditions. To answer this, we knocked down key enzymes in the fat body's PE and PC biosynthesis pathways and assessed HDF behavior in response to starvation when flies are fed NF (*Figure 7A*). Note that for the starvation experiments in these UAS/Gal4 conditions, we observed lethality on a 0% sucrose diet. Hence, we used a low-nutrient diet (1% sucrose agar) to induce nutrient deprivation (see 'Methods'). We found that knocking down the PC-biosynthesis enzyme Pyct2 did not affect HDF (*Figure 7A*). Similarly, the knockdown of eas, which initiates the early enzymatic reaction of PE biosynthesis (*Nyako et al., 2001*), did not influence HDF (*Figure 7A*). However, knocking down the rate-limiting enzymes in the ER (Pect) and mitochondria (PISD)-mediated PE biosynthesis pathways (*Zhao and Wang, 2020*) led to a diminished HDF response even on normal diets (*Figure 7A*). Our results are consistent with a critical role for PE enzymes in regulating HDF behavior even under a normal diet.

Interestingly, the knockdown of eas, PISD, and Pect all led to a loss of HDF on HSD treatment (*Figure 7*, lower panel). We measured HDF response in 14-day HSD-fed flies with fat body-specific Pect overexpression (Pect OE). These flies maintained a hunger response on HSD (*Figure 7B*). Since 14 days of HSD exposure is a critical timepoint for HDF breakdown in the wild-type flies (*Figure 1C and H*), this suggests that adipocyte Pect activity regulates hunger response. Consistent with our findings that 14-day HSD-fed flies show defective hunger response independent of TAG store levels (*Figure 1F, I and J*), we found that Pect knockdown does not alter baseline TAG store levels (*Figure 6—figure supplement 3*) but disrupts hunger response (*Figure 7A*). Together, these results show that Pect activity in the adult fly adipocytes is critical for regulating hunger response.

## Discussion

Several studies have shown a link between chronic sugar consumption and altered hunger perception (*Penaforte et al., 2013*; *Prinz, 2019*). Although the neuronal circuits governing hunger and HDF behavior have been well studied (*Lin et al., 2019*), less is known about the impact of adipose tissue dysfunction on feeding behavior. Using a *Drosophila* DIO model, we show that phospholipids, specifically PE, play a crucial role in maintaining HDF behavior.

The *Drosophila* model organism is a relevant model for human DIO and insulin resistance (*Kim et al., 2021*). Studies from Dus and Ja et al. have previously performed measurements on taste preference, feeding behavior/intake, survival, etc., using an HSD-induced obesity model, and have found much in common with their mammalian counterparts (*May et al., 2019*; *Deshpande et al., 2014*). However, the longest measurement of adult feeding behavior has been capped at 7 days (*van Dam*

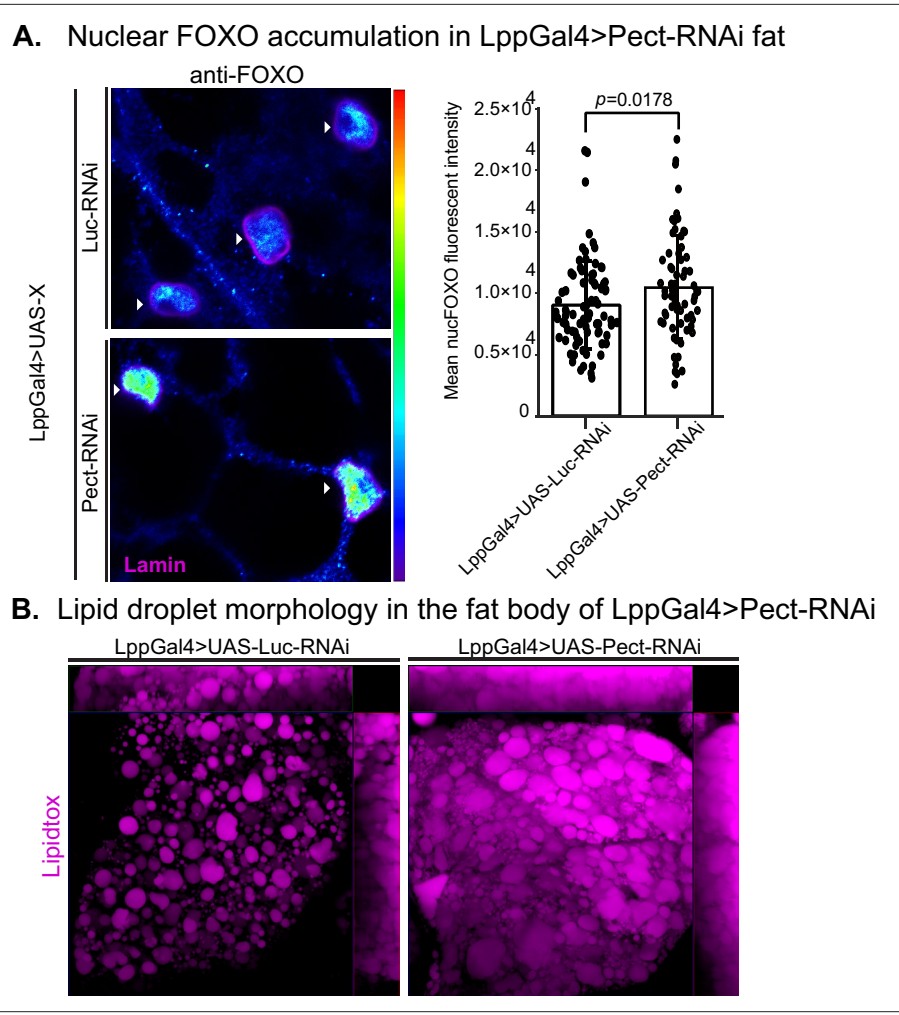

**Figure 5.** Pect knockdown in fly fat alters forkhead box O (FOXO) nuclear accumulation and lipid droplet morphology. (**A**, left) Representative confocal images of Lpp-Gal4>UAS-Luc-RNAi and Lpp-Gal4>UAS-Pect-RNAi fly fat immunostained with anti-FOXO antibody (blue to green, intensity-based) and lamin (pink) on day 14. Arrowhead points to the nucleus. (**A**, right) Mean nuclear FOXO fluorescent intensity of fat bodies from control (LppGal4>UAS-Luc-RNAi) and fat-specific Pect knockdown flies (LppGal4>UAS-Pect-RNAi). N = each circle represents a nucleus. All data were collected from a single experiment. Two-sided Wilcoxon rank-sum test. Error bars = standard deviation. Asterisks indicate significant changes with p-value<0.05, p-value<0.005, and p-value<0.0005. Error bars = standard deviation. (**B**) Representative confocal images of lipid droplets (magenta) in the fat bodies of control (LppGal4>UAS-Luc-RNAi) and fat-specific Pect knockdown flies (LppGal4>UAS-Pect-RNAi). Note that lipid droplet morphology in the fat-specific Pect knockdown flies resembles that of wild-type flies on high-sugar diet (HSD).

The online version of this article includes the following source data for figure 5:

**Source data 1.** Nuclear forkhead box O (FOXO) accumulation in LppGal4>Pect-RNAi fat.

---

et al., 2020). A recent study by Musselman and colleagues analyzed the fly lipidome on 3-week and 5-week HSD in a tissue-specific manner (**Tuthill et al., 2020**) and identified changes in neutral fat stores in the cardiac tissue (**Tuthill et al., 2020**).

In this study, we defined that a 14-day exposure of adult *Drosophila* to an HSD regime disrupts hunger response (**Figure 1C and H**). On evaluating HSD regime-induced lipid composition changes at this critical 14-day point, we uncovered a critical requirement for adipocyte PE homeostasis and a fat-specific role for the PE enzyme Pect in controlling HDF (**Figure 7**). Pect function in the adult fly adipocytes is critical for appropriate fat-to-brain lipoprotein delivery (**Figure 4**) and the maintenance of systemic insulin sensitivity (**Figure 5**). In sum, we identify that adipocyte-specific loss of

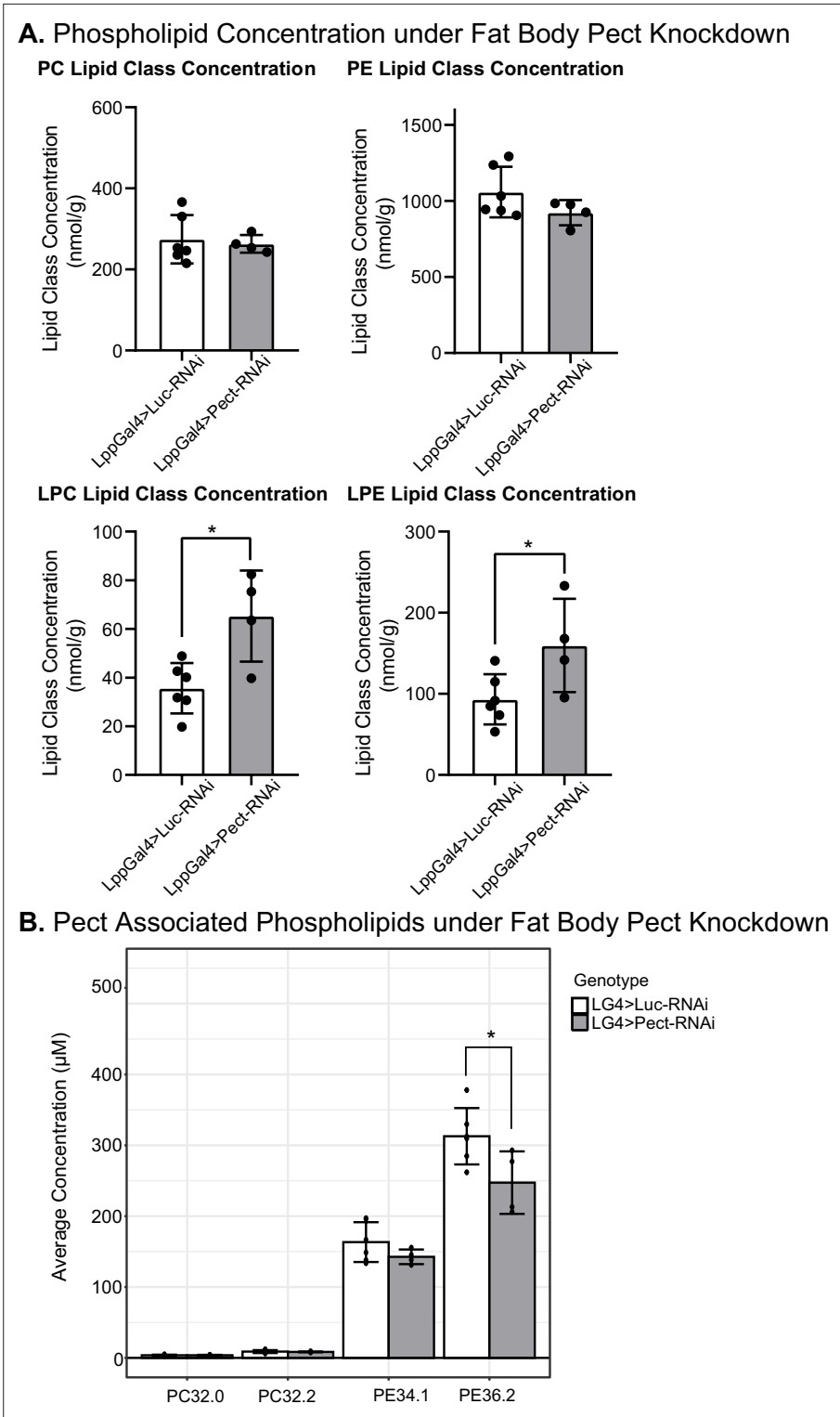

**Figure 6.** Manipulating Pect levels in the fat body alters phospholipid profile. (**A**) Average concentrations of phosphatidylcholine (PC), phosphatidylethanolamine (PE), lysophosphatidylcholine (LPC), and lysophosphatidylethanolamine (LPE) lipid classes in LppGal4>UAS-Pect-RNAi flies compared to control under 7-day normal food (NF) conditions. (**B**) Depicts the concentration of Pect-associated phospoholipids in LppGal4>UAS-Pect-RNAi flies compared to control. Lipidomics was performed using a targeted quantitative lipidyzer (Sciex 5500 Lipidyzer). 4–6 independent biological replicates were used for each genotype, with n = 10 flies composing one biological replicate. Unpaired *t*-test with Welch's correction. Asterisks indicate significant

*Figure 6 continued on next page*

*Figure 6 continued*

changes with p-value<0.05, p-value<0.005, and p-value<0.0005. Error bars = standard deviation, points = individual replicate values.

The online version of this article includes the following source data and figure supplement(s) for figure 6:

**Source data 1.** Lipidomics for fat-specific Pect knockdown and Pect overexpression on normal food.

**Figure supplement 1.** Pect mRNA expression.

**Figure supplement 1—source data 1.** Pect expression in RNAi and OE lines related to *Figure 6—figure supplement 1A*.

**Figure supplement 1—source data 2.** Pect expression in RNAi and OE lines related to *Figure 6—figure supplement 1B*.

**Figure supplement 2.** Fourteen days of high-sugar diet (HSD) cause an increase in Pect-associated phospholipid classes.

**Figure supplement 3.** Additional lipid class responses to fat body Pect knockdown.

**Figure supplement 4.** Lipidomic profile of fat body Pect overexpression flies.

Pect phenocopies the metabolic dysfunctions observed in a chronic HSD regime in adult flies. Therefore, we propose that PE homeostasis, specifically Pect activity in fat tissue, regulates HDF response (*Figure 8*).

## HSD leads to insulin resistance and a progressive loss of HDF behavior

Changes in feeding behavior in both vertebrates and invertebrates occur via communication between peripheral organs responsible for digestion/energy storage and the brain (*Prinz, 2019*). This communication is facilitated by factors that provide information on nutritional state (*Ahima and Antwi, 2008*). One example of such a factor is leptin, released from the adipose tissue and acts on neuronal circuits in the brain to promote satiety (*Rajan and Perrimon, 2012*; *Friedman and Halaas, 1998*). While leptin has long been studied as a satiety hormone, recent work in mice and flies suggests that a key function of leptin and its fly homolog upd2 regulates starvation response (*Brent and Rajan, 2020*; *Rajan and Perrimon, 2012*; *Ahima et al., 1996*; *Huang et al., 2020*). Indeed, we have previously shown that exposing flies to HSD alters synaptic contacts between Leptin/Upd2 sensing neurons and Insulin neurons. However, it resets within 5 days (*Brent and Rajan, 2020*), suggesting that yet-to-be-defined mechanisms maintain homeostasis on surplus HSDs beyond 5 days.

We analyzed feeding behavior over time to delineate how HSD alters the starvation response. We found that under normal diet conditions flies display a clear response to starvation in the form of elevated feeding that we termed 'hunger-driven feeding (HDF),' which was independent of age (*Figure 1B*). In contrast, chronic exposure to HSD led to a progressive loss of HDF that began on day 14 (*Figure 1C*). It could be argued that loss of HDF is simply due to an elevation of TAG storage in HSD-fed flies (*Figure 1—figure supplement 2*), thus losing the need to feed on starvation. However, several pieces of evidence support the idea that HSD affects feeding behavior independently of nutrient sensing. Under our experimental conditions, we find basal feeding to be statistically similar between NF-fed and HSD-fed conditions at all timepoints with the exception of day 10 (*Figure 1D*). Note that Dus and colleagues reported that on a 20% sucrose liquid diet for 7 days elevated food interactions (*May et al., 2019*, *May et al., 2020*). However, the Dus et al. studies are not comparable with our study due to the large differences in experimental protocol. They evaluated taste preference changes and feeding interactions on 5–30% sucrose liquid diet in 24-hr window over a period of 7 days. We assess food interaction in a 3-hr window, after providing a complex lab standard diet, to monitor HDF. Future studies would be needed to assess the effect of 14-day HSD on taste perception using the experimental design in this study (*Figure 1A*).The HDF response of HSD-fed flies (*Figure 1C*, days 3–10) is significantly lower than that of NF-fed flies, but they sense energy deficit and mobilize fat stores accordingly (*Figure 1F and G*). Hence, HSD-fed flies can calibrate their HDF to compensate only for the amount of fat lost in starvation. Nonetheless, this capacity of flies to couple energy sensing and feeding motivation is lost beyond day 14, as evidenced by the loss of HDF (*Figure 1C*, *Figure 1—figure supplement 1A*) and continuous TAG breakdown (*Figure 1F and G*). Strikingly, subjecting 14-day HSD-fed flies to prolonged starvation (up to 32 hr) was insufficient to induce increased HDF (*Figure 1H*). While there was an uptick in feeding behavior at 20 hr

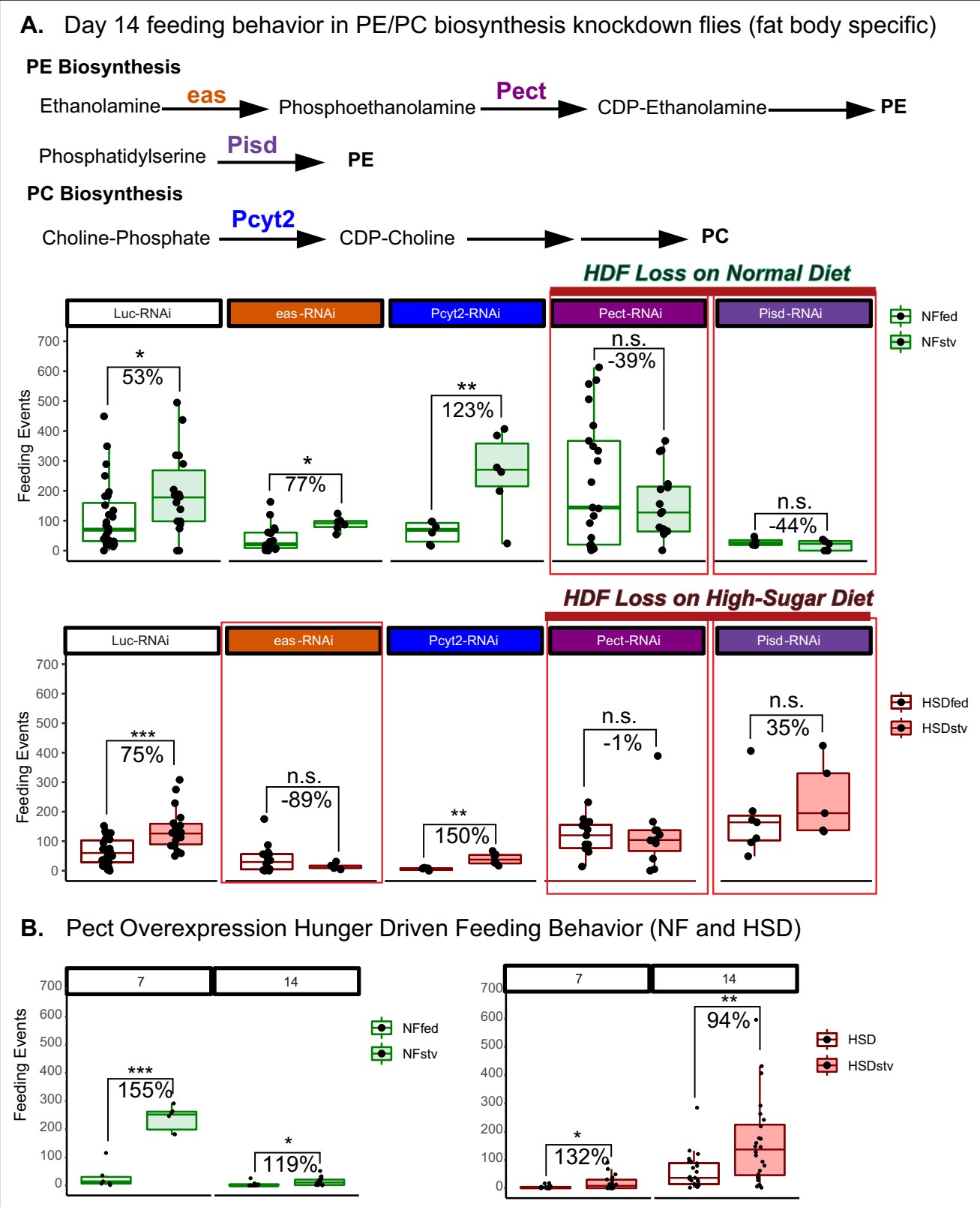

**Figure 7.** Manipulating Pect levels in the fat body alters hunger-driven feeding behavior. (**A**) A simplified schematic of the phosphatidylethanolamine (PE) and phosphatidylcholine (PC) biosynthesis pathway. eas and Pect are enzymes in the PE biosynthesis pathway, whereas PCYT2 is an enzyme in the PC biosynthesis pathway. (**A**, top) Average fly feeding events over time for normal food (NF) flies that were either fed or starved using a 1% sucrose agar diet (stv) for 16 hr prior to measurement. (**A**, bottom) Average feeding events over time for high-sugar diet (HSD)-fed flies that were either fed or starved

*Figure 7 continued on next page*

*Figure 7 continued*

for 16 hr prior to measurement using a 1% sucrose agar diet (stv). (**B**) Average feeding events of fed and starved Lpp-Gal4>Pect-overexpression on NF (left) and HSD (right) at 7 and 14 days of diet treatment. Each dot denotes an individual fly. Asterisks indicate significant changes with p-value<0.05, p-value<0.005, and p-value<0.0005. Error bars = standard deviation. For (**A**) and (**B**), each dot denotes an individual fly. Two-way ANOVA with Sidak post-test correction.

The online version of this article includes the following source data for figure 7:

**Source data 1.** Day 14 feeding behavior in phosphatidylethanolamine (PE)/phosphatidylcholine (PC) biosynthesis enzymes fat-specific knockdown flies relevant to *Figure 7A*.

**Source data 2.** Pect fat-specific overexpression of hunger-driven feeding behavior (normal food [NF] and high-sugar diet [HSD]) relevant to *Figure 7B*.

of starvation, this hunger response was not sustained at 24 and 32 hr (*Figure 1H*), even though flies continued to mobilize TAG reserves at 24 and 32 hr (*Figure 1I and J*). Thus, prolonged exposure to HSD leads to uncoupling nutrient sensing and feeding behavior.

Notably, fly and mammalian DIO models have striking differences and similarities. Mice show linear weight gain on obesogenic diets, but flies' rigid exoskeleton limits their capacity to store TAG beyond a certain point (*Han et al., 2020*; *Hatori et al., 2012*; *Lin et al., 2000*; *Figure 1—figure supplement 2*). However, similar to mammals, prolonged exposure to HSD, strongly associated with phospholipid dysregulation (*Sharma et al., 2013*; *Chang et al., 2019*; *Anjos et al., 2019*), leads to reduced insulin sensitivity (*Chang et al., 2019*; *Musselman et al., 2011*). We show that the levels of Dilp5, the fly's insulin ortholog, are reduced in the IPCs of HSD-fed flies (*Figure 2A*). However, we do not detect a decrease in Dilp5 or Dilp2 mRNA levels (*Figure 2—figure supplement 1*); this is suggestive of increased insulin secretion on HSD, similar to previously reported (*Pasco and Léopold, 2012*). Consistent with the idea that 14-day HSD triggers insulin resistance, we observe elevated FOXO nuclear localization in the fat bodies of the HSD-fed flies (*Figure 2B*), despite a likely increase in Dilp5 secretion on HSD (*Figure 2A*). Again, these findings align with mammalian studies showing that dysregulated FOXO signaling is implicated in insulin resistance, type 2 diabetes, and obesity (*Gross et al., 2008*).

## HSD and fat-specific Pect-KD cause changes to the phospholipid profile

Changes in the lipidome are strongly correlated with insulin resistance and obesity (*Mousa et al., 2019*). However, less is known about how the lipidome affects feeding behavior. To this end, we analyzed the lipid profiles of NF and HSD-fed flies over time (*Figure 3*). As expected, exposure to HSD increased the overall content of neutral lipids compared to the NF flies, with TAGs and DAGs increasing the most, which is consistent with other DIO models (*Musselman et al., 2011*; *May et al., 2019*; *Deshpande et al., 2014*). Surprisingly, we noted that 14 days of HSD treatment caused a decrease in FFAs and a rise in TAGs and DAGs (*Figure 3—figure supplement 1A*). We speculate that this reduction in FFA may be due to their involvement in TAG biogenesis (*Weiss et al., 1960*). We were interested to see whether the decrease in FFA correlated to a particular lipid species as PE and PC are made from DAGs with specific fatty acid chains. However, further analysis of FFAs at the species level did not reveal any distinct patterns. Most FFA chains decreased in HSD, including 12.0, 16.0, 16.1, 18.0, 18.1, and 18.2 (*Figure 3—figure supplement 1B*). This data was more suggestive of a global decrease in FFA, likely converted to TAG and DAG rather than depleting a specific fatty acid chain.

On day 14 of HSD treatment, when HDF response begins to degrade (*Figure 1C*, *Figure 1—figure supplement 1*), PE and PC levels rise dramatically, whereas LPE significantly decreases. Interestingly, similar patterns of phospholipid changes have been associated with diabetes, obesity, and insulin resistance in clinical studies (*Sharma et al., 2013*; *Chang et al., 2019*; *Anjos et al., 2019*; *Lim et al., 2011*), yet no causative relationship has been established (*Sharma et al., 2013*; *Chang et al., 2019*; *Anjos et al., 2019*). Intriguingly, we find that PC balance appears dispensable for maintaining HDF-response. But both the mitochondrial and cytosolic PE pathways seem critical for HDF response (*Figure 7*). Multiple pathways synthesize PE. Studies have shown that in addition to the mitochondrial PISD (*Steenbergen et al., 2005*) and cytosolic CDP-ethanolamine Kennedy pathway (*Calzada et al., 2016*; *Birner et al., 2001*), PE can be synthesized from LPE (*Riekhof and Voelker, 2006*). This pathway is named the exogenous lysolipid metabolism (ELM) pathway. ELM can substitute for the loss of the PISD pathway in yeast and requires the activity of the enzyme lyso-PE acyltransferase (LPEAT)

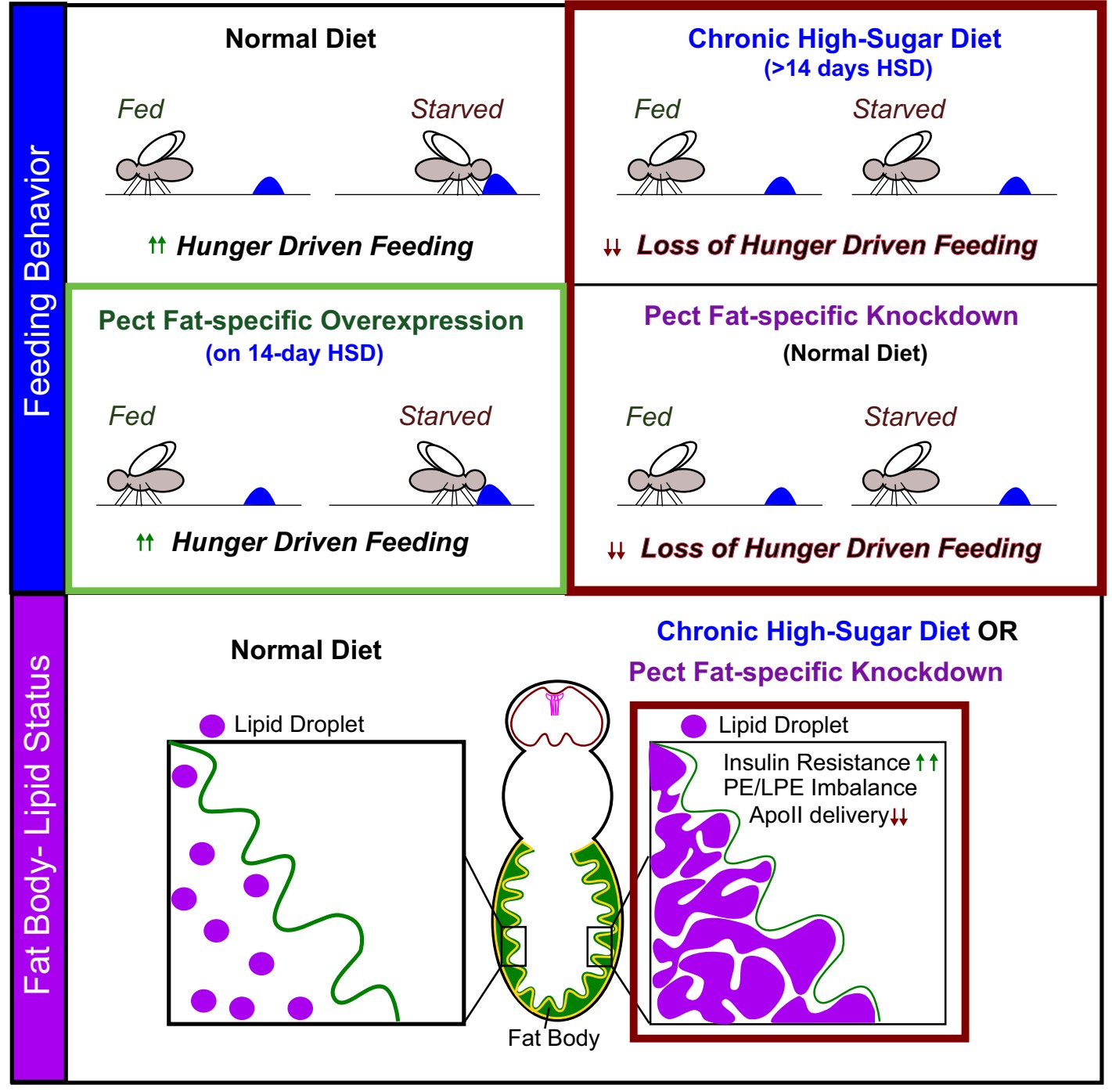

**Figure 8.** Pect function in adult fly adipocytes regulates hunger response. We identify that fat-specific knockdown of the rate-limiting phosphatidylethanolamine (PE) synthesis enzyme, Pect, *even on normal diet*, phenocopies the metabolic effects of subjecting flies to chronic (>14 days) high-sugar diet. We show that Pect function, in the adult fly adipocytes, is required for appropriate hunger response (top panel), insulin signaling, and fat to brain lipoprotein delivery (lower panel). We find that fat body-specific Pect overexpression can prolong appropriate hunger response on high-sugar diets (green box, upper panel).

that converts LPE to PE (*Riekhof et al., 2007*). In this study, we noted PE levels were upregulated on HSD while LPE levels were downregulated (*Figure 3*).

In contrast, fat-specific Pect-KD caused PE levels to trend downward, whereas LPE was upregulated (*Figure 6A*). Though the level changes for PE and LPE are contrasting between 14-day HSD

lipidome and Pect-KD, under both states, there is an imbalance of phospholipids classes PE and LPE. Hence, we propose maintaining the compositional balance of phospholipid classes PE and LPE is critical to HDF and insulin sensitivity.

The role of the minor phospholipid class LPE remains obscure. Our study observes that the LPE imbalance occurs during prolonged HSD exposure (*Figure 3*) and when fat body Pect activity is disrupted (*Figure 6A*). This suggests that LPE balance likely plays a role in insulin sensitivity and the regulation of feeding behavior. We anticipate that this observation will stimulate interest in studying this poorly understood minor phospholipid class. In future work, it would be interesting to test how the genetic interactions between the enzyme that converts LPE to PE, called LPEAT (*Riekhof et al., 2007*), and Pect manifest in HDF. Specifically, it will be interesting to ask whether reducing or increasing LPEAT will restore PE-LPE balance to improve the HDF response in HSD-fed flies and Pect-KD. Future studies should explore how LPE-PE balance can be manipulated to affect feeding behaviors.

In addition to changes in phospholipid classes (*Figure 6—figure supplement 2C*), we found that HSD caused an increase in the concentration of PE and PC species with double bonds (*Figure 3—figure supplement 1C*). Double bonds create kinks in the lipid bilayer, leading to increased lipid membrane fluidity, impacting vesicle budding, endocytosis, and molecular transport (*Ben M'barek et al., 2017*; *Choudhary et al., 2018*). Hence, a possible mechanism by which HSD induces changes to signaling by altering the membrane biophysical properties, such as by increased fluidity; this would impact various cellular processes, including synaptic firing and inter-organization vesicle transport. Consistent with this idea, we observe a significant reduction in the trafficking of ApoII-positive lipophorin particles from adipose tissue to the brain. Targeted experiments are required to understand how lipid membrane fluidity alters hunger response fully.

## Pect activity in fat and its impact on fat–brain communication

To explore the idea that fat–brain communication may be perturbed under HSD and Pect knockdown, we chose to examine a fat-specific signal known to travel to the brain. ApoLpp chaperones PE-rich vehicles called lipophorins traffic lipids from fat to all peripheral tissues, including the brain (*Palm et al., 2012*). ApoII, the Apolpp fragment harboring the lipid-binding domain, has been shown to regulate systemic insulin signaling by acting on a subset of neurons in the brain (*Brankatschk et al., 2014*). We found that both HSD treatment and Pect knockdown reduced ApoII levels in the brain (*Figure 4A and B*). Given that ApoII acts as a ligand for lipophorin receptors in the brain, ApoII may be a direct regulator of feeding. Alternatively, it could ferry signaling molecules and PE/PC lipids. In the future, it would be important to explore whether lipoprotein trafficking from fat-to-brain directly impacts the hunger response.

## Conclusion

We have uncovered a role for the phospholipid enzyme Pect as an important component in maintaining HDF. Future work should explore the precise mechanism of how Pect and the associated disruption in phospholipid homeostasis can impact adipose tissue signaling. In sum, this study lays the groundwork for further investigation into Pyct2/Pect as a potential therapeutic target for obesity and its associated comorbidities.

## Methods
### Animals used and rearing conditions

The following strains were used in this article: w1118, PISD-RNAi (Bloomington #67763), Pect-RNAi (Bloomington #67765), eas-RNAi (Bloomington #38528), Pcyt2-RNAi (Bloomington #67764), Lpp-Gal4 on X (P.Leopold/S. Eaton [*Géminard et al., 2009*]), Luciferase-RNAi (FlyBase ID: JF01355), and w;;UAS-Pect III (FlyBase ID:FBal0347227, generously donated by Clandinin lab), UAS-HA-Apolopp-myc (generously donated by S. Eaton) (*Brankatschk and Eaton, 2010*). Flies were housed in 25°C incubators. In all experiments, only adult male flies were used. Flies were sexed upon eclosion and place on normal diet, a standard diet containing 15 g yeast, 8.6 g soy flour, 63 g corn flour, 5 g agar, 5 g malt, 74 mL corn syrup per liter, for 7 days. Anesthesia using a $CO_2$ bubbler was used for initial sexing, then never used for the remainder of the experiment. After 7 days, flies were either maintained on normal diet or moved to an HSD, composed of the same composition as normal diet but with an additional

300 g of sucrose per liter (30% increase) for the length specified in the figures (typically 7 or 14 days). For measurements of HDF, a portion of flies from each diet were placed on starvation media (0% sucrose/1% agar) for 16 hr prior to the experiment.

## Immunostaining

Immunostaining of adult brains and fat bodies was performed as previously described (*Brent and Rajan, 2020*; *Rajan et al., 2017*). Tissues were dissected in ice-cold PBS. Brains were fixed overnight in 0.8% paraformaldehyde (PFA) in PBS at 4°C, and fat bodies were fixed in 4% formaldehyde in PBS at room temperature. Following fixation, tissues were washed five times in 0.5% BSA and 0.5% Triton X-100 in PBS (PAT). Tissues were pre-blocked in PAT + 5% NDS for 2 hr at room temperature, then incubated overnight with the primary antibody in block at 4°C. Following incubation, tissues were washed five times in PAT, re-blocked for 30 min, then incubated in secondary antibody in block for 4 hr at room temperature. Samples were washed five times in PAT, then mounted on slides in Slow fade gold antifade. Primary antibodies were as follows: chicken anti-Dilp2 (1:250; this study); rabbit anti-Dilp5 (1:500; this study); rabbit anti-ApoII (1:500; this study), rabbit anti-FOXO (1:500, gift from Leopold Pierre); and mouse-anti-lamin (1:100; ADL67.10 DSHB; RRID:AB_528336). Secondary antibodies from Jackson ImmunoResearch (1:500) include donkey anti-rabbit Alexa 647 (RRID:AB_2492288); donkey anti-rabbit Alexa 488 (RRID:AB_2313584); donkey anti-chicken Alexa 647 (RRID:AB_2340379); and donkey anti-rabbit Alexa 594 (RRID:AB_2340621). Lipid droplets were stained with lipidtox (1:500, Thermo Fisher Cat#H34477) overnight at room temperature. Images were captured with Zeiss LSM 800 confocal system and analyzed with ImageJ.

## Image analysis

Quantification of the number of puncta and percentage area occupied by ApoII immunofluorescence was done using ImageJ. Maximum-intensity projections of z-stacks that spanned the entire depth of the IPCs at 0.3 um intervals were generated. A region of interest was manually drawn around the IPCs and a binary mask for the ApoII channel was created using automated Moments thresholding values, which was followed by watershed postprocessing to separate particles. The number of particles and the area fraction were measured using the 'analyze particles' function.

All images were acquired on the same day to minimize variability among samples. The fluorescent intensity of Dilp 5 was measured using z-stack summation projections that included the full depth of the IPCs. A region of interest around the IPCs was manually drawn and the integrated density values were acquired using ImageJ (*Schneider et al., 2012*). To measure nuclear FOXO accumulation, a similar number of confocal stacks were acquired for each tissue sample. For FOXO, image analysis was performed in the MATLAB R2020b environment, and the associated scripts are available at GitHub address. FOXO accumulation in *Drosophila* adult fat cells was assessed by measuring the mean voxel GFP intensity in the nucleus that is delimited by the lamin antibody signal. To generate the 3D nuclear masks, lamin stacks were first maximally projected along the z-axis, and after global thresholding, basic morphological operations, and watershed transforms, the locations of the nuclear centroids in the x-y plane were used to scan the z-stacks and reconstruct the nuclear volume. The accuracy of the segmentation was assessed by manual inspection of random cells. Once the nuclear compartment was reconstructed in 3D, it was used as a volumetric mask to extract intensity values of the FOXO reporter signal and compute the mean voxel intensity in the nucleus. Two-sided Wilcoxon rank-sum tests were performed to assess the statistical significance of pairwise comparisons between experimental conditions.

## Lipidomics

Whole adult male flies were flash frozen in liquid nitrogen either after 7 or 14 days on normal diet or HSD. 10 flies were used per biological sample and 10 biological replicates were used for each diet and timepoint. For Pect manipulation lipidomics, LppGal4>UAS-Luc-RNAi, LppGal4>UAS-Luc, LppGal4>UAS-Pect-RNAi, and LppGal4>UAS-Pect flies were generated and flash frozen after 7 days on NF. Frozen samples were sent to the Northwest Metabolomics Research Center for targeted quantitative lipid profiling using the Sciex 5500 Lipidyzer (see *Hanson et al., 2019* for detailed methods).

## Feeding behavior

For HDF analysis, age-matched w1118 flies were given a normal diet or HSD for 5, 7, 14, 21, 24, and 28 days after an initial 7 days of development on a normal diet. All other experiments were performed for 7-day or 14-day durations. Then, 16 hr prior to feeding behavior assessment, half of the flies from each treatment were moved to starvation media 0%. During the 3-hr assessment window of feeding behavior, individual flies were placed in a single well of FLIC and supplied with a 5% sucrose liquid diet for all FLIC experiments. For feeding behavior involving Gal4>UAS manipulation, a 1% sucrose agar diet was substituted for the 0% sucrose agar 16 hr starvation to avoid fly death. Detailed methods for how FLIC operates can be found in *Ro et al., 2014*. Fly feeding was measured for the first 3 hr in the FLIC, and all FLICs were performed at 10 am local time. For each FLIC, half of the wells (n = 6/FLIC) contained the fed group, and the other half contained the starved group of flies for direct comparison. A total of 12–30 flies were measured for analysis of feeding. Any signal above 40 (a.u.) was considered a feeding event. Analysis of feeding events was performed using R.

## Triglyceride measurement

Whole-body TAG measurements were performed in accordance with previously published methods (*Rajan et al., 2017*). In brief, whole flies (n = 3) were used per biological replicate with 8–9 replicates used for each timepoint and treatment. Flies were collected for TAG for all timepoints and treatments measured for feeding behavior (see above). Data was normalized to TAG levels per number of flies. Significance was calculated using two-way ANOVA.

## Survival assay

Survival curves were performed using flies harvested in a 24-hr time frame and aged for 7 days, then subjected to either normal lab food or HSD. Ten males per vial were flipped onto 1% sucrose agar starvation food. The number of dead flies was recorded each day until every fly had died. Flies were kept in a 25°C incubator with a 12 hr light–dark cycle for the entirety of the experiment. Survival analysis was performed using the Survival Curve module of GraphPad Prism. A Mantel–Cox test was used to determine statistical significance. Greater than 90 flies were used per condition per curve.

## Gene expression

Thirty fly fat bodies (for Pect) or heads (for Dilps) of each genotype were dissected in RNAlater. Immediately after dissection, fat bodies were moved to tubes of 200 µL RNAlater on ice. RNAlater was then removed, 30 µL of trireagent and a scoop of beads were added, and fat bodies were homogenized using a bullet blender. RNA was then isolated using a Direct-zol RNA microprep kit following the manufacturer's instructions. Isolated RNA was synthesized into cDNA using the Bio-Rad iScript RT supermix for RT-qPCR and qPCR was performed using the Bio-Rad ssoAdvanced SYBR green master mix. Primers used in the article are as follows: Robl (endogenous control), forward: AGCG GTAGTGTCTGCCGTGT and reverse: CCAGCGTGGATTTGACCGGA; Pect, forward: CTGGAAAA GGCTAAGAAACTGGG and reverse: TCTTCAGTGACACAGTAGGGAG; alpha-tubulin (endogenous control), forward: ATCGAACTTGTGGTCCAGACG and reverse: GGTGCCTGGAGGTGATTTGG ; Dilp2, forward: GCCTTGATGGACATGCTGA and reverse: CATAATCGAATAGGCCCAAGG; Dilp5, forward: GCTCCGAATCTCACCACATGAA and reverse: GGAAAAGGAACACGATTTGCG; Pect. Relative quantification of mRNA was performed using the comparative CT method and normalized to Robl mRNA expression. For each experiment, three biological replicates were used with three technical replicates used for qPCR.

## Statistical methods

A *t*-test or ANOVA was performed using PRISM. Boxplots and standard deviation calculations were either performed through Prism or R. R packages used in this article included tidyverse (http://dx. doi.org/10.21105/joss.01686), ggplot2 (https://ggplot2.tidyverse.org), and ggthemes (https://github. com/jrnold/ggthemes, *Arnold, 2022*).

## Acknowledgements

We thank Dr. Pierre Leopold for generously donating the FOXO antibody used in this article. We are grateful to Dr. Thomas Clandinin for gifting the UAS-PECT transgenic flies and the late Dr. Susan Eaton for the generous gift of the Lpp-Gal4 flies. We would also like to thank the Northwest Metabolomics Research Center team for their support in lipidomic profiling. This work was possible due to grants awarded to AR from NIGMS (GM124593) and New Development funds from Fred Hutch Cancer Center. KPK was supported by the NIH Chromosome Metabolism and Cancer Training Grant (T32CA009657) and is currently supported by the NSF Post-Doctoral Research Fellowship (NSF Award #2109398). Genomic reagents from the DGRC funded by NIH grant 2P40OD010949 were used in this study. Stocks obtained from the Bloomington Drosophila Stock Center (NIH P40OD018537) and Transgenic RNAi Resource Project (NIGMS R01 GM084947 and NIGMS P41 GM132087) were used in this study. We are immensely grateful to the three anonymous peer reviewers for their incisive, constructive, and balanced feedback that significantly contributed to improving the clarity of this study.

## Additional information

### Funding

| Funder | Grant reference number | Author |
| --- | --- | --- |
| National Institute of General Medical Sciences | GM124593 | Akhila Rajan |
| Directorate for Biological Sciences | 2109398 | Kevin P Kelly |
| NIH Chromosome Metabolism and Cancer Training Grant | T32CA009657 | Kevin P Kelly |

The funders had no role in study design, data collection and interpretation, or the decision to submit the work for publication.

### Author contributions

Kevin P Kelly, Formal analysis, Investigation, Visualization, Methodology, Writing – original draft; Mroj Alassaf, Data curation, Formal analysis, Writing – original draft, Writing – review and editing; Camille E Sullivan, Ava E Brent, Zachary H Goldberg, Data curation, Formal analysis, Investigation; Michelle E Poling, Investigation; Julien Dubrulle, Visualization; Akhila Rajan, Conceptualization, Supervision, Funding acquisition, Writing – review and editing

### Author ORCIDs

Kevin P Kelly http://orcid.org/0000-0002-8632-2339
Mroj Alassaf http://orcid.org/0000-0001-6277-9417
Zachary H Goldberg http://orcid.org/0000-0003-2972-9682
Julien Dubrulle http://orcid.org/0000-0002-4186-7749
Akhila Rajan http://orcid.org/0000-0003-1160-5796

### Decision letter and Author response

Decision letter https://doi.org/10.7554/eLife.80282.sa1
Author response https://doi.org/10.7554/eLife.80282.sa2

## Additional files

### Supplementary files

• MDAR checklist

### Data availability

All data generated or analysed during this study are included in the manuscript and supporting file; Source Data files have been provided for Figures.

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
