## [Editor Report]

This manuscript posits that genetically-induced misregulation of phospholipid levels in fat cells causes defective hunger-driven feeding behaviors in adult *Drosophila melanogaster*. In parallel, the study also presents the rescue of feeding defects observed in diet-induced obese flies via the overexpression of the rate-limiting enzyme of phospholipid synthesis (PECT) in fly fat. This is an important paper that presents evidence of a potential causative relationship between phospholipid dysregulation and satiety sensing. This work will be of interest to a broad group of metabolism, obesity, and feeding behavior researchers.

---

## [Decision Letter]

[Editors' note: this paper was reviewed by Review Commons.]

Thank you for submitting "Fat Body Phospholipid State Dictates Hunger Driven Feeding Behavior" for consideration at *eLife*. Your article has been reviewed by the same 3 peer reviewers that initially reviewed your work at Review Commons, and the evaluation has been overseen by a Reviewing Editor and a Senior Editor.

Comments to the Authors:

The reviewers were very positive and reassured by the lipidomics data clarifying the relationship between Pect levels, phospholipid levels, and HSD. However, the major concern remains: the manuscript does not have sufficient support for the model that hunger signals are indeed disrupted by Pect manipulations. An alternative hypothesis is still possible, that fatter flies have delayed hunger perception due to the extra levels of stored energy and, therefore, a delayed need for food, and would need to be starved longer in order to become hungry. For this reason, we are sorry to say that, after consultation with the reviewers, we have decided that at this time this work will not be considered for publication by *eLife*. However, due to the excitement about a potential molecular mechanism of hunger response, if the major concern can be addressed, we would be happy to reconsider this decision.

Specifically, one of the experimental suggestions from one of the reviewers is "to test a range of fasting times. Knowing that the authors' have affected 'hunger driven feeding' regardless of fasting time would be impressive. I would also suggest doing a simple starvation resistance experiment." If results show that indeed there is no difference in the hunger-driven response and that it's all due to differences in energy stores, we would not be as excited to reconsider the work. If, on the other hand, levels of energy stores are uncoupled from this response, we would be enthusiastic to revisit this work.

[Editors’ note: further revisions were suggested prior to acceptance, as described below.]

Thank you for resubmitting your work entitled "Fat Body Phospholipid State Dictates Hunger Driven Feeding Behavior" for further consideration by *eLife*. Your revised article has been evaluated by David James (Senior Editor) and a Reviewing Editor.

The manuscript has been greatly improved but there are some remaining minor issues that need to be addressed in the text, as outlined below:

1) The new starvation survival data is inconsistent with previous literature showing that higher sugar diets lead to fatter flies and increased starvation resistance. This could be due to differences in the experimental setup (survival on 1% sucrose/agar versus agar alone). This experiment is therefore not easy to interpret since it is not clear how much of the 1% sucrose the flies are consuming. We suggest that you remove Figure S1B and the corresponding text since the data should not be interpreted as showing that HSD flies are "starvation" sensitive, as they have access to 1% sucrose.

2) Note that for the hunger/feeding measurements after "starvation", the Methods state that sometimes flies were starved for 16 hr on "0% sucrose liquid diet" (is this just water?- please clarify), and sometimes they were "starved" for 16 hr on "1% sucrose". Please ensure these different approaches are clear when presenting the data and the data are interpreted in the right context (of full starvation (0% sucrose liquid diet) or low-calorie sugar diet (1% sucrose).

3) Please cite PMID: 31067455 and PMID: 32539934, AND discuss your data in the context of their findings on how high sugar diets affect sweet taste perception and satiation. While the manuscript is cited, we'd like to have you include in the discussion how their results (e.g. 7 days on HSD significantly increased feeding via the FLIC assay, the same technique used in this manuscript) fits with what you describe here. Specifically discuss, why you think you don't see this same increase in feeding with your chronic HSD?

4) The feeding assay used in this manuscript- FLIC assay- does not actually measure food intake. Rather, it measures how often the fly touches the food. PMID: 31067455 and PMID: 32539934 called them "licks" in their manuscript. Since it is well established from several labs (e.g., Dus, Neely) that chronic HSD decreases taste sensitivity to sugar. We suggest adding this caveat to the Discussion: "It's not clear how food interactions (and therefore measurements using the FLIC assay) are affected by sweet taste sensitivity, which is known to be altered by chronic HSD. Future studies might…"

---

## [Author Response]

We are very grateful for this reviewer identifying this this methods description error and bring it to our attention. We used 0% sucrose agar for overnight starvation in this study as most labs do. The error occurred because we were using another manuscript from the lab to help draft the methods section (PMID: 29017032). In that study, where we assayed the effect of *chronic starvation* our lab used: “*1% sucrose agar for 5 days at 25C”*. However, in this current study, because we are testing *acute effects of overnight starvation*, we are using 0% sucrose agar.

Pect mRNA level is higher with HSD. This is surprising because not only, as authors mention, is increased PC32.2 with HSD suggests lower Pect activity, but also because Pect RNAi phenocopies long-term HSD in HDF behavior, lipid morphology, FOXO accumulation in fat body. The authors speculate that the data "likely shown an upregulation in an attempt to mediate the Pect dysregulation occurring at the protein level." If that were true, a western blot may be informative. Zhao and Wang (2020, PLoS Genetics) generated a Pect antibody that seems compatible with western blot applications. That being said, I don't think such data is critical for the manuscript. I mention this simply as a suggestion for the authors.

a. page 8, line 22-23, did you mean to write "Given how PC32.2 is elevated after 14 days of exposure to HSD, we assumed that Pect levels would be low for flies under HSD," not "high?" Otherwise the subsequent 2 sentences don't make sense.

We agree that the most confusing aspect of the study was that Pect mRNA levels being very high on Day 14 HSD, but nonetheless the effects of Pect-KD phenocopied HSD. To resolve this, we have now performed lipidomic analyses on whole adult flies, when Pect is knocked-down (KD) by RNAi in the fat tissue. We now present a new dataset in Figure 6.

Two striking changes occur. They are:

Pect-KD shows increase in the phospholipid classes LPC and LPE (Figure 6A). In contrast, LPE is significantly downregulated on HSD Day 14 (Figure 3).Pect-KD shows a significant reduction in specific class of PE 36.2 (Figure 6B). Our data regarding increase in PE 36.2 agree with a previous lipidomic analyses of Pect mutant retina (PMID: 30737130). In contrast, PE 36.2 trends upwards on 14 day HSD (Figure S7C) though not significantly.

On 14-day HSD consistent with extreme upregulation of Pect mRNA fed flies (Figure S6A; Pect mRNA 200-250 fold), PE trends upwards on 14-day HSD (Figure 3) and PE 36.2 trends higher (Figure S7C). We note that on the surface of it PE and LPE *per se* are contrasting between 14day HSD lipidome and fat-specifc Pect-KD. But there is a significant commonality that under both states there is an imbalance of phospholipids classes PE and LPE. Hence, we propose that maintaining the compositional balance of phospholipid classes PE and LPE is critical to hunger-driven feeding and insulin sensitivity. Hence, either increase or decrease, of these key phospholipid species, may lead to abnormal hunger-driven feeding.

We agree that a western blot would be informative as well, but we were unable to obtain the reagent from Dr. Wang’s group, precluding us from performing this request.

To ensure that we appropriately discuss and clarify this issue, we have now included a section in the discussion – Page 14 Lines 26-34- under the subtitle “*The implications of relationship between Pect levels and HSD*”. We have pasted an excerpt from that subsection below for this reviewers assessment.

“Also, we note that over-expression of Pect cDNA in the fat-body does not alter phospholipid balance (Figure S9) and indeed improves HDF on HSD (Figure 7B). While this may appear inconsistent, it is critical to note that over-expression of Pect cDNA using UAS/Gal4 only increases Pect mRNA expression by 7-fold (Figure S6A), whereas HSD causes its upregulation by 250-fold (Figure S6B). Hence, we speculate that an increased ‘basal’ level of Pect such as by that provided by a cDNA over-expression in fat, may be protective to the negative effects of HSD (Figure 7B) without affecting overall phospholipid levels (Figure S9), but extreme upregulation Pect on HSD affects the PE and LPE balance (Figure 3).”

Reviewer #1 (Significance (Required)):The work is potentially novel and interesting, but at this stage it's difficult to interpret what the phenotype signifies. Although the manuscript could be revised simply by modifying the text, experimentally addressing the concerns would significantly improve the work.

In sum, we hope we have addressed the key concern for Reviewer #1 as to whether the behavior we report here is indeed a dampening of starvation-induced feeding, or an effect of increase in baseline feeding. We hope that by reviewing our non-normalized data, they can appreciate that it is the former. Also, we hope that Reviewer #1 appreciates that we have strived to address the concerns by additional experiments, to clarify our findings and improve the impact of the work.

Reviewer #2 (Evidence, reproducibility and clarity (Required)):This intriguing manuscript by Kelly and colleagues uses the fruit fly *Drosophila* melanogaster as a model to understand how diet-induced obesity alters the feeding response over time. In particular, the authors findings indicate that chronic exposure to a high-sugar diet significantly alters the starvation-induced feeding response. These behavioral studies are complemented by a lipidomics approach that reveals how a chronic high sugar affects many lipid species, including phospholipids. The authors then pursue mechanistic studies that indicate phospholipid metabolism within the fat body appears to remotely affect insulin secretion from the insulin producing cells. Moreover, the changes in phospholipid abundance are associated with changes in insulin-signaling, including increased insulin secretion from the IPCs and elevated levels of FOXO within the nucleus.I find the study to be potentially very important – the authors combine a longitudinal study that would be difficult in any other model with the powerful genetic tools available in the fly. The conclusions are mostly convincing, but a few follow-up experiments are required:

We are grateful for the reviewers constructive, detail-oriented, and balanced feedback, and their recognition of the value of this study. Now, we have performed additional experiments to address the key concerns raised by all reviewers. We hope that on reading the revised version of our study, that the reviewer continues to feel positive about the message of this study and its potential impact.

1. The key conclusions from the manuscript assume that manipulation of Pect expression levels alters phosphatidylethanolamine (PE) levels. However, the authors make no attempt to verify that the genetic experiments described herein actually affect PE levels. At a minimum, changes in PE levels should be verified for the Pect knockdown and overexpression lines. Similarly, there is no evidence that manipulation of either EAS or Pcyt2 induces the expected metabolic effects. I'm not asking that the longitudinal feeding experiments be repeated, simply that the authors measure the relevant lipid species, preferably with a targeted LC-MS approach.

Prompted by this reviewer, we performed targeted LC-MS on whole adult flies, on normal diet, to assess lipid levels for fat-specific Pect-KD and overexpression. We decided to focus on Pect, as its knock-down even on normal diet causes a dampened hunger-driven feeding behavior (Figure 7A) and phenocopied a 14-day HSD feeding phenotype.

We now present a new dataset in Figure 6. Two striking changes occur:

They are:

i) Pect-KD shows a significant reduction in specific class of PE 36.2 (Figure 6B). Our data regarding decrease in PE 36.2 agree with a previous lipidomic analyses of Pect mutant retina (PMID: 30737130). It is to be noted that though overall levels of all PE species trend downwards, like the Clandinin lab study on Pect (PMID: 30737130), we did not find a significant change in the overall PC and PE levels.

ii) Pect-KD shows increase in the phospholipid classes LPC and LPE (Figure 6A). In contrast, LPE is significantly downregulated on HSD Day 14 (Figure 3).

On 14-day HSD consistent with extreme upregulation of Pect mRNA fed flies (Figure S6A; Pect mRNA 200-250 fold), PE trends upwards on 14-day HSD (Figure 3) and PE 36.2 trends higher (Figure S7C). We note that on the surface of it PE and LPE *per se* are contrasting between 14day HSD lipidome and fat-specifc Pect-KD. But there is a significant commonality that under both states there is an imbalance of phospholipids classes PE and LPE. Hence, we propose that maintaining the compositional balance of phospholipid classes PE and LPE is critical to hunger-driven feeding and insulin sensitivity. Hence, either increase or decrease, of these key phospholipid species, may lead to abnormal hunger-driven feeding.

Finally, fat-specific Pect-OE did not cause significant changes to lipid species (Figure S9). This could either be due to the fact that in fat-specific Pect-OE flies under normal food and that we were assaying whole body lipid levels and not fat-specific lipid changes. But to counter that, even a 60% reduction in Pect mRNA levels (Figure S6A), was sufficient to produce an effect on whole body phospholipid balance (Figure 6). Hence, we speculate that by maintaining a basally higher (7-fold higher Pect mRNA level Figure S6A), might allow 14-day HSD-fed flies to buffer the negative effects of HSD and we predict that it might take longer to disrupt the phospholipid balance and HDF response.

We have now included a section in the discussion – Page 14 Lines 26-34- under the subtitle “The implications of relationship between Pect levels and HSD”. We have pasted an excerpt from that subsection below for this reviewers assessment.

“Also, we note that over-expression of Pect cDNA in the fat-body does not alter phospholipid balance (Figure S9) and indeed improves HDF on HSD (Figure 7B). While this may appear inconsistent, it is critical to note that over-expression of Pect cDNA using UAS/Gal4 only increases Pect mRNA expression by 7-fold (Figure S6A), whereas HSD causes its upregulation by 250-fold (Figure S6B). Hence, we speculate that an increased ‘basal’ level of Pect such as by that provided by a cDNA over-expression in fat, may be protective to the negative effects of HSD (Figure 7B) without affecting overall phospholipid levels (Figure S9), but extreme upregulation Pect on HSD affects the PE and LPE balance (Figure 3).”

A central hypothesis in the study is that the HSD over a period of 14 days results in insulin resistant and that these changes are leading to changes in hunger dependent feeding. I would encourage the authors to determine if Foxo mutants are resistant to these HSD-induced effects on HFD.

We thank the reviewers for this suggestion. However, given that dFOXO nuclear localization rather than expression levels regulate insulin sensitivity, we feel that disrupting dFOXO levels via mutation or knockdown will produce a plethora of indirect effects including developmental abnormalities (PMID: 24778227, PMID: 16179433, PMID: 29180716, PMID: 12893776). Our data suggest that chronic HSD treatment and Pect affect insulin sensitivity in fat tissue. However, we feel that investigating whether insulin sensitivity/FOXO signaling in fat tissue regulates feeding behavior is outside the scope of our work.

In lines 25-30, the authors draw the conclusion that an increase in unsaturated fatty acid species is associated with the HSD and that these changes results in a more fluid lipid environment. While I agree with the model, the manuscript contains no evidence to support such a model. Either test the hypothesis or move the last line of the section to the discussion.

We thank the reviewer for this important and insightful comment. We agree that the data we presented and discussed in the original version is at the moment speculative. Addressing the hypothesis that increase in unsaturated fatty acid species result in a more fluid lipid environment will require us to build tools and expertise. Hence, this hypothesis is better suited for exploration in a future study. Given this, we have moved this out of the Results section into the Discussion section titled “*HSD and fat-specific PECT-KD causes changes to phospholipid profile*” (See excerpt below from page 13, lines 24-35).

“In addition to changes in phospholipid classes, we found that HSD caused an increase in the concentration of PE and PC species with double bonds (Figure S4C and S4D). Double bonds create kinks in the lipid bilayer, leading to increased lipid membrane fluidity which impacts vesicle budding, endocytosis, and molecular transport^14,92^. Hence it is possible that a mechanism by which HSD induces changes to signaling is by altering the membrane biophysical properties, such as by increased fluidity, which would have a significant impact on numerous biological processes including synaptic firing and inter-organ vesicle transport.”

Also, as per the reviewer’s guidance, given that we are speculating here, we have also shifted this dataset from Main figure 4 to supplement S4C and S4D.

In addition, lines 25-30 state that FFAs are increased after 14 days of a HSD. Figure 3A shows the exact opposite – FFAs are significantly decreased in 14 day fed animals despite being elevated in the 7 day fed animals. This is an interesting result that warrants discussion. Moreover, I would encourage to examine the lipidomic data more carefully to ensure that the text accurately portrays the lipid profiles.

We apologize for misstating that FFAs are decreased on 14-day HSD in the lines 25-30. It was an error and we have corrected this. We agree with the reviewer that the reduction of FFA on Day 14-HSD is an intriguing and unexpected observation that needs to be emphasized and further discussed. To this end, we have added figure S4B, wherein we have provided the difference in FFA concentration (by species) after days 7 and 14.

Furthermore, we have discussed what the potential meaning of reduced FFA at Day 14 implies in page 12, lines 19-27 of the Discussion section titled “*HSD and fat-specific PECT-KD causes changes to phospholipid profile*”. We have stated the following-

“We speculate that this reduction in FFA maybe due to their involvement in TAG biogenesis (PMID: 13843753). We were interested to see if the decrease in FFA correlated to a particular lipid species, as PE and PC are made from DAGs with specific fatty acid chains. However, further analysis of FFAs at the species level did not reveal any distinct patterns. The majority of FFA chains decreased in HSD, including 12.0, 16.0, 16.1, 18.0, 18.1, and 18.2 (Figure S4B). This data was more suggestive of a global decrease in FFA, likely being converted to TAG and DAG, rather than a specific fatty acid chain being depleted.”

The processed lipidomics data should also be included as supplementary data table so that they can be independently analyzed by the reader.

We thank the reviewer for this suggestion. As per the reviewers request, we have included the raw data as an attachment in our supplementary material (Supplementary Files 1-3.), so that interested readers can use the datasets generated in this study for future work and further analysis.

*Page 3, Line 1 and 2: "…have been shown to impact feeding behavior and metabolism that leads to…" This is an awkward and grammatically incorrect sentence.*

*Page 3, Lines 7-32 is one very large paragraph but contains concepts that should be broken down over at least three paragraphs.*

*Page 3, Line 25: A description of the reaction catalyzed by Pect would be helpful for a manuscript focused on Pecte activity.*

*Page 4, Line 10: "previously characterized method of eliciting diet induced feeding behavior." As stated in the text, the method is previously described yet the manuscript characterizing the method isn't cited.*

*Figure legend 3 contains a random assortment of capitalized lipid species. Also, the names of lipid species are inappropriately broken into multiple names. Please use correct nomenclature throughout the manuscript.*

The list above is nowhere near comprehensive. The manuscript requires significant editing.

We are grateful to the reviewer for drawing our attention to these errors. We have made significant edits to the revised manuscript to address the above-mentioned concerns, as well as made additional textual changes throughout and copyedited it. We hope that the reviewer will find the manuscript reads better and the clarity and preciseness is significantly improved.

Reviewer #2 (Significance (Required)):I find the study to be potentially very important – the authors combine a longitudinal study that would be difficult in any other model with the powerful genetic tools available in the fly. The findings will significantly advance our understanding of how lipid metabolism links dietary nutrition with feeding behavior.

Once again, we are grateful for this reviewer’s thoughtful critique and encouraging words regarding our work and its potential impact.

Reviewer #3 (Evidence, reproducibility and clarity (Required)):Summary:This manuscript uses *Drosophila* to investigate how diet-induced obesity and the changes in the lipid metabolism of the fat boy modulate hunger-driven feeding (HDF) response. The authors first demonstrate that chronic exposure (14 days) of high sugar diet (HSD) suppresses HDF response. Through lipidome analysis, the authors identify a specific class of lipids to be elevated upon chronic HSD feeding. This coincided with the changes in expression of Pect, an enzyme that regulates the biosynthesis of these lipids. Modulating the expression of Pect specifically in the fat body affected HDF response.

We thank this reviewer for their rigorous and thoughtful critique and for identifying a key issue with our original study pertaining to a gap in how Pect mRNA levels on 14-day HSD are elevated but the Pect-KD phenocopies the HDF. Now by performing whole-body adult fly lipidomic on fat-specific Pect-KD we have resolved this issue and provided clarity on role of Pect in maintaining phospholipid homeostasis and thus subsequently impacts hunger-driven feeding. We hope the reviewer finds that the revised manuscript provides further clarity to the functional link between Pect’s role in fat-body and hunger-driven feeding.

Major comments:The author claim that the HDF response in HSD is distinct between early (5d, 7d) and chronic (day 14) HSD feeding. However, the data seem to indicate that HDF response is significantly decreased at all time points in HSD. For example, at day 5 HDF response was increased only 3fold in HSD (Figure 1C) compared to around 50-fold increase in NF (Figure 1B). The scale of the Y-axis in Figure 1B and 1C is an order of magnitude different. Including the starved data (NFstv and HSDstv) in Figure S1, normalized to NF fed group, would better visualize the overall trends. Related to this, having the source data for the actual number of feeding events would be useful (e.g., to see the baseline changes in feeding in different time points in Figure 1 and the effect of genetic manipulations in Figure 7).

As per the reviewers request, we now have modified our graphs to show source data (Figure S1) and show the raw feeding events.

Then in the non-normalized graphs we plot, over a longitudinal time course, baseline and hunger-driven feeding events (Figure 1B-D). We also show that HSD fed flies do not display increased baseline feeding (Figure 1D) suggesting that the effect we see on HDF are no clouded by increased baseline feeding.

Yes, the reviewer makes an important point that HDF response on HSD fed flies is of a lower magnitude than NF fed flies. We think that is a biologically meaningful observation, as it suggests that flies have a remarkably fine-tuned ability to coordinate food-intake with nutrient store levels.

Now we have included a paragraph in the Discussion, Page 11 Lines 23-27, that say the following to ensure the readers appreciate this salient point raised by this reviewer.

It is to be noted that the HDF response of HSD-fed flies (Figure 1C, Days 3-10) is of lower order of magnitude than the NF-fed flies. This suggests that in addition to sensing an energy deficit and mobilizing fat stores (Figure 1F, 1G, S1), HSD fed flies calibrate their starvation-induced feeding to compensate only for the lost amount of fat. Overall, this suggests that flies have a remarkably finetuned ability to coordinate food-intake with nutrient store levels.The association between fat body Pect level and phospholipid levels is not clear. Day 14 of HSD feeding shows high expression of Pect in the fat body and elevated levels of PC32.0 and PC32.2. The authors assume the high expression of Pect in the fat body is due to the compensatory response, but there are no data indicating downregulation of Pect levels at the earlier time points of HSD feeding. A previous study demonstrated that Pect mutant flies have lower levels of PC32.0 but higher PC32.2 (PMID: 30737130).

We agree that one puzzling aspect of the original version of this study was that Pect mRNA levels being very high on Day 14 HSD, but nonetheless the effects of Pect-KD phenocopied HSD. To resolve this, prompted by Reviewer #2 and #3 concerns, for this revised version we have now performed lipidomic analyses on whole adult flies, when Pect is knocked down (KD) by RNAi in the fat tissue. We now present a new dataset in Figure 6. Two striking changes occu. They are:

Pect-KD shows increase in the phospholipid classes LPC and LPE (Figure 6A). In contrast, LPE is significantly downregulated on HSD Day 14 (Figure 3).Pect-KD shows a significant reduction in specific class of PE 36.2 (Figure 6B). Our data regarding increase in PE 36.2 agree with a previous lipidomic analyses of Pect mutant retina (PMID: 30737130). In contrast, PE 36.2 trends upwards on 14 day HSD (Figure S7C) though not significantly.

On 14-day HSD consistent with extreme upregulation of Pect mRNA fed flies (Figure S6A; Pect mRNA 200-250 fold), PE trends upwards on 14-day HSD (Figure 3) and PE 36.2 trends higher (Figure S7C). We note that on the surface of it PE and LPE *per se* are contrasting between 14day HSD lipidome and fat-specifc Pect-KD. But there is a significant commonality that under both states there is an imbalance of phospholipids classes PE and LPE. Hence, we propose that maintaining the compositional balance of phospholipid classes PE and LPE is critical to hunger-driven feeding and insulin sensitivity. Hence, either increase or decrease, of these key phospholipid species, may lead to abnormal hunger-driven feeding.

On day 14, HDF response was increased 70-fold in w1118 flies in NF (Figure 1B; w1118), but only 2.5-fold in lpp>LucRNAi control flies in NF (Figure 7A). This suggests that lpp-gal4 driver lines have a significant effect on HDF response. Using a different fat-body specific Gal4 line would be necessary to validate conclusions.

Regards reduced HDF magnitude, in our experience using UAS-Gal4 reduces HDF response magnitude consistently and cannot be compared to *w1118* which is more robust. To account for background differences, we use Uas-Gal4 with control RNAi. It clearly shows differences in HDF response on starvation, but Pect and Pisd RNAi does not (Figure 7A). Hence, given that this experiment internally controls for any changes in HDF response for UAS-Gal4>RNAi, we conclude that HDF response in disrupted in Pect and PISD KD (Figure 7).

We only presented the Lpp-driver in our study, as this driver is the only fat-specific driver that has no leaky expression in other tissues, and is specific to fat as apolpp promoter used to generate this Gal4 line is only expressed in fat tissue (Eaton and colleagues, PMID: 22844248). Other widely used fat-specific drivers, including the pumpless-Gal4 (*ppl-Gal4*) driver has leaky expression in gut or other tissues (See Table 2 of this detailed study by Dr. Drummond- Barbosa https://www.ncbi.nlm.nih.gov/pmc/articles/PMC7642949/). If the reviewer is aware of a fat-specific Gal4 line, other than Lpp-Gal4, which has a highly specific expression in the fat tissue without leaky expression in other tissues, then we are happy to take onboard the reviewer’s suggestion and try that fat-specific Gal4 that they suggest.

HSD feeding promotes Pect expression (Figure S3C) and global changes in phospholipid levels (Figure 3, 4). Therefore, shouldn't Pect overexpression (not Pect RNAi) in a normal diet mimic HSD feeding state and promote loss of HDF response? Conversely shouldn't knockdown of Pect in HSD rescue loss of HDF response?

We agree that a puzzling aspect is that Pect mRNA levels are significantly elevated in HSD Day-14, but Pect-KD showed displays the inappropriate HDF response. As we have described in our response to this reviewer on Page 19, we believe that Pect-KD and HSD disrupt PE and LPE balance overall but in different ways. Whereas Pect-OE using cDNA expression in fat body does not cause a significant change to any lipid class (Figure S9), and our results suggest that basally higher level of PECT is likely to be protective on HSD with respect to HDF (Figure 7B).

To ensure that we appropriately discuss and clarify this issue, we have now included a section in the discussion – Page 14 Lines 26-33- under the subtitle “The implications of relationship between Pect levels and HSD”. We have pasted an excerpt from that subsection below for this reviewers assessment. 5 “Also, we note that over-expression of Pect cDNA in the fat-body does not alter phospholipid balance (Figure S9) and indeed improves HDF on HSD (Figure 7B). While this may appear inconsistent, it is critical to note that over-expression of Pect cDNA using UAS/Gal4 only increases Pect mRNA expression by 7-fold (Figure S6A), whereas HSD causes its upregulation by 250-fold (Figure S6B). Hence, we speculate that an increased ‘basal’ level of Pect such as by that provided by a cDNA over-expression in fat, may be protective to the negative effects of HSD (Figure 7B) without affecting overall phospholipid levels (Figure S9), but extreme upregulation Pect on HSD affects the PE and LPE balance (Figure 3).”

We would have liked to test Pect protein expression on HSD, but since we were unable to access antibodies for Pect published in a prior study (PMID: 33064773) from Dr. Wang’s lab (see Page 10-11, of response to Reviewer #1). Hence, we were unable to test how the proteins levels of Pect correlate with the 250-fold increase mRNA expression.

In conclusion, we hope the reviewer appreciates that our results regarding Pect function are consistent with the main conclusion that achieving the right phospholipid balance between PE and LPE, is critical for an organism to display an appropriate HDF response.

Minor comments:All graphs should plot individual data points and showed as box and whisker plot as much as possible.

Thanks for this suggestion, we have added individual data points to the vast majority of figures in the paper. We have made exceptions to graphs such as seen in figure 1 and FigureS4B-D where we find individual data points add an unnecessary layer of complexity. We hope these changes provide additional clarity and strength to the claims made in this manuscript.

Data for day 14 missing in Figure S4A and S4B.

We have provided Day 14 for the PC composition and PE composition, due to changes in Figures, they are now S7A and S7B.

Reviewer #3 (Significance (Required)):The interactions between diet-induced obesity, peripheral tissue homeostasis and feeding behavior is an interesting topic that can be addressed using *Drosophila*. This manuscript demonstrates how fat body Pect levels affect HSD induced changes in hunger-driven feeding response. However, at this point, the functional association between fat body Pect level, global phospholipid level, and loss of hunger-driven feeding response in chronic HSD feeding is not clear.

We hope the revised data, and discussion of the paper, provides well-substantiated functional association on the importance of maintaining phospholipid balance, driven by Pect enzyme, as a critical regulator of hunger-driven feeding behavior. As stated in the revised discussion, the key take home message of our manuscript is that on prolonged HSD exposure PC, PE and LPE levels are dysregulated, the loss of phospholipid homeostasis coincided with a loss of hunger driven feeding. Following this lead on phospholipid imbalance, we then uncovered a critical requirement for the activity of the rate-limiting PE enzyme PECT within the fat tissue in controlling hunger-driven feeding.

[Editors’ note: what follows is the authors’ response to the second round of review.]

Comments to the Authors:The reviewers were very positive and reassured by the lipidomics data clarifying the relationship between Pect levels, phospholipid levels, and HSD. However, the major concern remains: the manuscript does not have sufficient support for the model that hunger signals are indeed disrupted by Pect manipulations. An alternative hypothesis is still possible, that fatter flies have delayed hunger perception due to the extra levels of stored energy and, therefore, a delayed need for food, and would need to be starved longer in order to become hungry. For this reason, we are sorry to say that, after consultation with the reviewers, we have decided that at this time this work will not be considered for publication by eLife. However, due to the excitement about a potential molecular mechanism of hunger response, if the major concern can be addressed, we would be happy to reconsider this decision.Specifically, one of the experimental suggestions from one of the reviewers is "to test a range of fasting times. Knowing that the authors' have affected 'hunger driven feeding' regardless of fasting time would be impressive. I would also suggest doing a simple starvation resistance experiment." If results show that indeed there is no difference in the hunger-driven response and that it's all due to differences in energy stores, we would not be as excited to reconsider the work. If, on the other hand, levels of energy stores are uncoupled from this response, we would be enthusiastic to revisit this work.

We thank you for reviewing our manuscript #13-05-2022-RA-RC-*eLife*-80282, titled "Fat Body Phospholipid State Dictates Hunger Driven Feeding Behavior". We much appreciate your time and efforts in consulting with the peer-reviewers and providing us with thoughtful comments on our submitted work.

We were pleased to note that the reviewers were reassured about the lipidomic datasets we presented in the revision. We noted that one unresolved issue in our interpretation of the hunger-driven feeding motivation of 14-day HSD-fed flies precluded the acceptance of our manuscript.

To recap, we had shown that by 16 hours of starvation, age-matched Normal Fed (NF) controls to display a strong hunger-driven feeding (HDF) response. But, the 14-day high sugar diet (HSD) fed flies do not. Figure S1. We interpreted this dataset to mean that prolonged 14-day exposure to HSD altered the hunger perception of flies on starvation.

However, the reviewers raised an alternative hypothesis: " fatter flies have delayed hunger perception due to the extra levels of stored energy and, therefore, a delayed need for food, and would need to be starved longer to become hungry".

To resolve this, the reviewer suggested two specific experiments.

"to test a range of fasting times. Knowing that the authors' have affected 'hunger driven feeding' regardless of fasting time would be impressive.""I would also suggest doing a simple starvation resistance experiment."

We have now performed these experiments and present our results below.

14-day HSD-fed flies were subject to HSD, then starved for 16, 20,24, and 32 hours*. We assayed the hungerdriven feeding of all the starvation time points against the baseline 14-day HSD-fed in the same FLIC run (Author response image 1). We observed that at 20 hours after starvation, flies display a short burst of feeding activity (Author response image 1). However, by 24 hours and 32 hours, they do not show increased feeding activity (Author response image 1). At 24 and 32 hours of starvation, these flies broke down 50% of their fat stores, which is statistically significant (Author response image 1, 1C). NF-fed flies break down 50% of their fat stores by the 16hour time point, sufficient for them to display hunger-driven feeding (See Figure 1 of our manuscript). Hence, even though 14-day HSD-fed flies use up a significant portion of their energy stores at 24-32 hours, they do not display hunger-driven feeding. A linear regression analysis between feeding events and TAG levels suggests that TAG store levels and feeding events do not show any significant correlation (Author response image 1). Hence, subjecting flies to a 14-day HSD disrupts their hunger-driven feeding regardless of fasting time and TAG stores.

* We find that beyond 32hrs starvation on a 0% sucrose diet, the flies die, so this is the furthest we can technically run this starvation response curve.

**Author response image 1. sa2fig1:** (**A**) Feeding events of 14-day HSD-fed flies at various starvation times. (**B**) TAG stores of 14-day HSD-fed flies at various starvation times. (**C**) Normalized TAG of 14-day HSD-fed flies at various starvation times. (**D**) Regression analysis of feeding events and TAG.

2.We performed a starvation survival curve (on 1% sucrose agar) on flies fed 14-day HSD and compared them to age-matched controls kept on 14-day NF. 14-day HSD-fed flies were significantly less resilient to starvation than NF-fed flies (Author response image 2). Again, this is in keeping with the disrupted hunger response of the HSD-starved flies. Note 70-100 flies were used for this experiment.

**Author response image 2. sa2fig2:** Starvation survival on 1% sucrose agar of 14-day NF vs. HSD fed flies.

We were puzzled by your note that "the manuscript does not have sufficient support for the model that hunger signals are indeed disrupted by Pect manipulations". Thus, we would like to draw your attention again to the results we presented in the manuscript on the role of PECT in hunger-driven feeding (Figure 7A). Please note that fatspecific knockdown of PECT in flies fed a "normal" diet is sufficient to reduce HDF (Figure 7A) significantly. Despite showing a disrupted hunger response (Figure 7A), we noted that PECT-KD does not cause a significant change in TAG stores on a normal diet (Author response image 3). Hence, PECT-KD in fat disrupts hunger response in flies on normal lab diets without impacting energy store levels.

**Author response image 3. sa2fig3:** PECT-KD in fat tissue on normal diets does not alter TAG stores.

[Editors’ note: further revisions were suggested prior to acceptance, as described below.]

The manuscript has been greatly improved but there are some remaining minor issues that need to be addressed in the text, as outlined below:1) The new starvation survival data is inconsistent with previous literature showing that higher sugar diets lead to fatter flies and increased starvation resistance. This could be due to differences in the experimental setup (survival on 1% sucrose/agar versus agar alone). This experiment is therefore not easy to interpret since it is not clear how much of the 1% sucrose the flies are consuming. We suggest that you remove Figure S1B and the corresponding text since the data should not be interpreted as showing that HSD flies are "starvation" sensitive, as they have access to 1% sucrose.

We note your concern on the 0% vs 1% sucrose diets. 1. In standard lab conditions, flies live only a short 32-36 hours on 0% sucrose diets. In our experience, evaluating these short 0% starvation curves are unreliable. Hence, we and others have used 1% sucrose diets to analyze starvation response curves (PMID: 29017032; 23021220), since they allow flies to live for up to 15 days. Since the original email did not specify that these curves must be performed on non-standard 0% diet. we performed what is standard in the field for starvation curves i.e., 1% sucrose diet. Again, we feel that inclusion of this data is important, and is in response to a reviewer request. But given your reservations, we have removed the figure 1- supplement 1B and references to it in results and discussion. We note that this change has no significant impact on the conclusions of our study.

2) Note that for the hunger/feeding measurements after "starvation", the Methods state that sometimes flies were starved for 16 hr on "0% sucrose liquid diet" (is this just water?- please clarify), and sometimes they were "starved" for 16 hr on "1% sucrose". Please ensure these different approaches are clear when presenting the data and the data are interpreted in the right context (of full starvation (0% sucrose liquid diet) or low-calorie sugar diet (1% sucrose).

We have now included a more detailed methods section, as follows to describe the starvation conditions.

For hunger-driven feeding analysis, age-matched *w1118* flies were given a normal diet or HSD for 5, 7, 14, 21, 24, and 28 days after an initial 7 days of development on a normal diet. All other experiments were performed for 7-day or 14-day durations. 16 hours prior to feeding behavior assessment, half of the flies from each treatment were moved to starvation media 0%. During the three hour assessment window of feeding behavior, Individual flies were placed in a single well of fly liquid-food interaction counter (FLIC) and supplied with a 5% sucrose liquid diet for all FLIC experiments. For feeding behavior involving Gal4>UAS manipulation, a 1% sucrose agar diet was substituted for the 0% sucrose agar 16 hours starvation to avoid fly death. Detailed methods for how FLIC operates can be found in Ro et al., 201447. Fly feeding was measured for the first three hours in the FLIC and all FLICs were performed at 10 am local time. For each FLIC, half of the wells (n=6/FLIC) contained the fed group, and the other half contained the starved group of flies for direct comparison. 12-30 flies were measured for analysis of feeding. Any signal above 40 (a.u.) was considered a feeding event. Analysis of feeding events was performed using R.

We note that only for datasets in figure 7, given that 0% sucrose agar diet caused lethality, we used the 1% sucrose agar diet. We have clearly indicated this – 1% sucrose diet- in the legends and Results section Page 8, lines 16-18.

“Note that for the starvation experiments in these UAS/Gal4 conditions, we observed lethality on a 0% sucrose diet. Hence, we used a low-nutrient diet (1% sucrose agar) to induce nutrient deprivation (See Methods).”

Also, in the revised version we specify that in the FLIC counter we use 5% sucrose liquid food during the feeding assessment for all experiments.

3) Please cite PMID: 31067455 and PMID: 32539934, AND discuss your data in the context of their findings on how high sugar diets affect sweet taste perception and satiation. While the manuscript is cited, we'd like to have you include in the discussion how their results (e.g. 7 days on HSD significantly increased feeding via the FLIC assay, the same technique used in this manuscript) fits with what you describe here. Specifically discuss, why you think you don't see this same increase in feeding with your chronic HSD?

The studies by Dus and colleagues evaluated food interactions in a 5-30% sucrose liquid diet over a 24-hour window for 7 days using FLIC. In our studies, we maintain flies on a standard lab (solid diet- which is more complex with proteins), or on 30% HSD (that is the addition of additional sucrose to the standard lab diet). We then evaluate their feeding interactions for 3 -hours only, after they have been starved or baseline. In Figure 1D, we presented results showing that under our conditions, 3-hour FLIC window, we did not observe a statistical significant change in baseline feeding behavior of NF vs. HSD fed flies within the 3-hour window. Nonetheless, our results don’t directly contradict the Dus et al. studies, given they are using a liquid sugar-based diet over a period of 7 days to measure feeding interactions. We have now discussed this as follows, in page 10 Line 15-19.

Under our experimental conditions, we find basal feeding to be statistically similar between NF fed and HSD fed conditions at all timepoints with exception of day 10 (Figure 1D). Note that Dus and colleagues reported that on a 20% sucrose liquid diet for 7-days elevated food interactions^44, 85^. However, the Dus et al. studies are not comparable with our study due to the large differences in experimental protocol. They evaluated taste preference changes and feeding interactions on 5-30% sucrose liquid diet in 24-hour window over a period of 7 days. We assess food interaction in a 3-hour window, after providing a complex lab standard diet, to monitor hunger-driven feeding.

4) The feeding assay used in this manuscript- FLIC assay- does not actually measure food intake. Rather, it measures how often the fly touches the food. PMID: 31067455 and PMID: 32539934 called them "licks" in their manuscript. Since it is well established from several labs (e.g., Dus, Neely) that chronic HSD decreases taste sensitivity to sugar. We suggest adding this caveat to the Discussion: "It's not clear how food interactions (and therefore measurements using the FLIC assay) are affected by sweet taste sensitivity, which is known to be altered by chronic HSD. Future studies might…"

As noted in response to #3 above, we have discussed the two studies and provided information to the readers that the experimental design of our work does not directly evaluate the taste sensitivity to sugar.

We note now that in the Discussion section in page 10 line 20 that:

Future studies would be needed to assess the effect of 14 day HSD on taste perception using the experimental design in this study (Figure 1A).